# Unraveling the optical shape of snow

Alvaro Robledano [1,2] ✉, Ghislain Picard[1], Marie Dumont [2], Frédéric Flin [2], Laurent Arnaud[1] & Quentin Libois [3]

The reflection of sunlight off the snow is a major driver of the Earth's climate. This reflection is governed by the shape and arrangement of ice crystals at the micrometer scale, called snow microstructure. However, snow optical models overlook the complexity of this microstructure by using simple shapes, and mainly spheres. The use of these various shapes leads to large uncertainties in climate modeling, which could reach 1.2 K in global air temperature. Here, we accurately simulate light propagation in three-dimensional images of natural snow at the micrometer scale, revealing the optical shape of snow. This optical shape is neither spherical nor close to the other idealized shapes commonly used in models. Instead, it more closely approximates a collection of convex particles without symmetry. Besides providing a more realistic representation of snow in the visible and near-infrared spectral region (400 to 1400 nm), this breakthrough can be directly used in climate models, reducing by 3 the uncertainties in global air temperature related to the optical shape of snow.

Ice crystals formed in the atmosphere show a large variety of sophisticated and, often, near-perfect geometric shapes[1] (Fig. 1a). The interaction of sunlight with such crystals sometimes results in well-known optical phenomena, called halos[2], whose nature is directly related to the shape of the crystals[3].

Once the ice crystals are deposited on the ground and form the snow cover, they establish bonds with each other and their shape drastically changes. As a consequence, the original atmospheric crystals are rarely distinguishable after a few hours or days[4,5], making snow cover a two-phase porous material made of ice and air, rather than a collection of individual particles. Snow is continuously evolving due to the thermodynamical non-equilibrium between ice and interstitial air that leads to recrystallisation of the water molecules, a process called metamorphism. This process leads to a variety of snow morphologies at the micrometer scale, called microstructure hereafter, which are less regular, less symmetrical and more diverse than the original crystals (Fig. 1b–d). Hence the interaction of sunlight with snow on the ground is more complicated to model than with ice clouds in the atmosphere, and is not yet fully understood. The resulting uncertainties significantly alter the estimation of the solar radiation reflected by snow and in turn the Earth's radiative budget[6,7]. Recent climate simulations thus show that simply changing the snow microstructure can modify the global annual-mean 2 m air temperature by nearly 1.2 K[8].

Early studies have tried to relate the snow optical properties to the snow microstructure, relying on strong approximations such as considering snow as a collection of disconnected and independent ice spheres[9]. Under this hypothesis, the Mie theory provides the optical properties for a given sphere size. Later on, the relative success of the equivalent-sphere concept, i.e., the representation of snow as a collection of spheres with the same volume-to-area ratio, to simulate snow albedo, led to the spread of this concept in most snow optical models[10–12]. This representation provided a clear relationship between the snow albedo and the sphere size[10,13], called the "optical diameter". However, posterior studies highlighted several caveats of the spherical assumption[14–16], particularly in determining how the snow reflects light in different directions and how deep the light penetrates[17–20]. This uncertain representation has implications for the interpretation of satellite data[21,22], for snow photochemistry[23–25], for light transmission through snow over sea ice[26], and more importantly for the surface energy budget of snow-covered regions[6–8]. This raises the need to go beyond the unrealistic but still widely used spherical representation of snow.

Many alternative shapes have been used to describe the snow microstructure, such as fractals[27], Voronoi tessellations[28], cubes, hexaedra and hyperboloids, and combinations thereof[16,29]. However, all these attempts have in common that they still represent snow as a

[1]Univ. Grenoble Alpes, CNRS, INRAE, IRD, Grenoble INP, IGE, Grenoble, France. [2]Univ. Grenoble Alpes, Université de Toulouse, Météo-France, CNRS, CNRM, Centre d'Etudes de la Neige, Grenoble, France. [3]CNRM, Université de Toulouse, Météo-France, CNRS, Toulouse, France. ✉ e-mail: alvaro.robledano-perez@univ-grenoble-alpes.fr

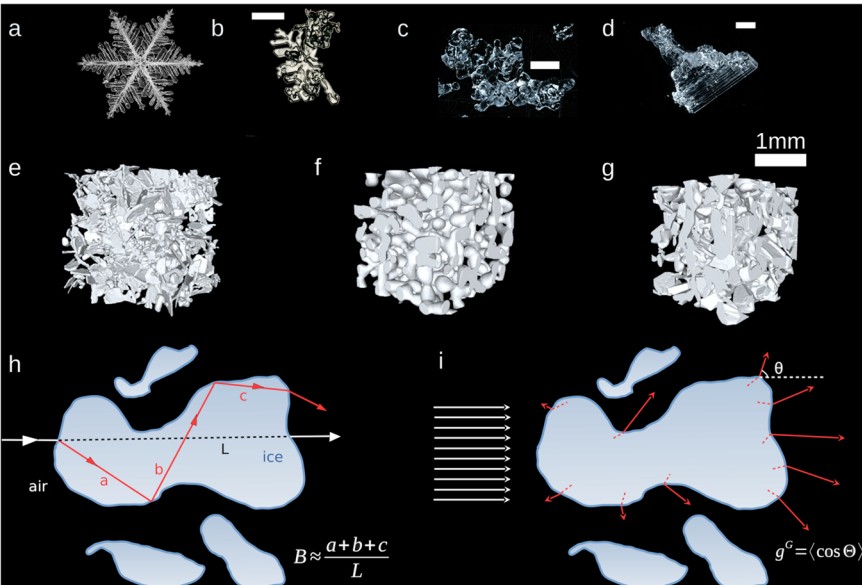

**Fig. 1 | From ice crystals to snow microstructure. a** Photograph of an ice crystal in the atmosphere (stellar dendrite - K. Libbrecht). **b** Photograph of natural snow on the ground (decomposing and fragmented precipitation particles - Météo-France). White scale bars: 1 mm. **c**, **d** Photographs of natural snow on the ground (rounded grains and depth hoar - F. Dominé). White scale bars: 1 mm. **e**–**g** 3D microstructure of three natural snow samples as revealed by X-ray tomography. From left to right: precipitation particles, rounded grains, and depth hoar. **h**, **i** Illustration of the absorption enhancement parameter $B$ and the geometric asymmetry parameter $g^G$ in a 2D space. In **h**, the red arrows represent a photon path accounting for refraction and internal reflections, while the white arrows represent propagation in a straight line. In **i**, the white arrows represent an incoming set of photons, and the red arrows represent an ensemble of possible outgoing photon paths.

collection of particles with well-defined shapes. Conversely, Malinka[30] considered snow as a two-phase random medium without any particular shape, but it is still unclear whether this approach is fully valid for representing the microstructure of natural snow[31,32]. Despite all these attempts to derive the optical properties of snow by fitting various shapes or hypotheses onto the complexity of the snow microstructure, little is known about its actual role. In simpler words, the "optical shape" of snow–a concept we introduce here following that of the "optical diameter"–remains largely unknown.

Here, we address the question of the optical shape of snow, across the wide diversity of snow microstructures. To do so, we apply a ray-tracing model to simulate the path and attenuation of light in 3D images issued from micro-computed tomography[33–35], which over two decades has provided detailed knowledge of the microstructure of snow with a resolution of a few micrometers (Fig. 1e–g). From the simulation results we deduce the values of two main optical shape parameters, namely the absorption enhancement parameter $B$ and the geometric asymmetry parameter $g^{G\,27}$ (Fig. 1h, i). $B$ quantifies the lengthening of the light path within the absorbing phase (the ice) due to refraction and internal reflections, and is important for accurately estimating the light absorption in snow. $g^G$ quantifies the tendency of the medium to scatter light forward or backward, and is essential to predict how deep light can penetrate into snow, or conversely how easily it is reflected back to the atmosphere. Both parameters characterize the optical shape independently of the size. Snow optical models, such as those implemented in climate models[36–38], directly or indirectly rely on prescribed values of these parameters, which are most of the time set to the values for spheres. $B$ and $g^G$ have previously been estimated for individual ice particles with particular shapes[27,39–42] or indirectly estimated from macroscopic measurements on snow[43,44]. However, direct estimation of $B$ and $g^G$ for natural snow along with their variability is still unknown[45]. In the present study, our simulations provide an accurate estimate of the range of these fundamental parameters for natural snow, paving the way for a more realistic representation of snow in optical and climate models.

## Results

### The optical shape of natural snow

The shape parameters $B$ and $g^G$ were computed over 33 snow microstructure images, that cover most of the snow types referenced in the international classification of seasonal snow on the ground[4] (see Supplementary Table 1). The computations are made with the Rough Surface Ray-Tracing (RSRT) model[46,47], which has been extended to simulate light propagation in 3D microstructure images (see Methods). For every image, simulations are run in the visible and near-infrared (NIR) spectral region, every 50 nm from 400 nm to 1400 nm. The simulations track $10^6$ photons through the microstructure and report the energy reflected back and the profile of energy within the snowpack, which are then used to deduce $B$ and $g^G$ (macroscopic method hereinafter). In addition, we implemented a microscopic approach (called geometric method) that records the traveled distance within the ice, and the direction changes between entering and exiting the ice (see Methods).

The parameters $B$ and $g^G$ of all snow samples at 900 nm are presented in Fig. 2, along with previous estimates obtained for idealized geometric shapes. Figure 2 also shows the values predicted by the two-phase random medium theory[30], where $B$ is related to the ice refractive index $n$ by $B = n^2$. $g^G$ can also be expressed in terms of $n$, in a less trivial way (see Supplementary Methods 1). We use here the 900 nm results for a comparison with previous studies[40,48]. These results are as well a compromise between the lower wavelengths where the modeling uncertainties are higher (Supplementary Fig. 1) and the higher wavelengths where the assumption of low absorption of ice is less valid than at shorter wavelengths. The absorption enhancement parameter $B$ of natural snow clearly clusters around 1.7 (mean of the 33 samples ± 1 standard deviation: 1.68 ± 0.02 in Fig. 2a, 1.70 ± 0.00 in Fig. 2c), while for idealized shapes it spans a larger range between 1.25 for spheres to 1.84 for fractals. However, except fractals, all other featured shapes have a $B$ smaller than 1.7, meaning that natural snow absorbs energy more efficiently than these idealized shapes. The value of 1.7 is comparable to that experimentally retrieved in[43] (1.6 ± 0.1), and is slightly higher than the one in[45] (1.49). It matches the predicted value for the

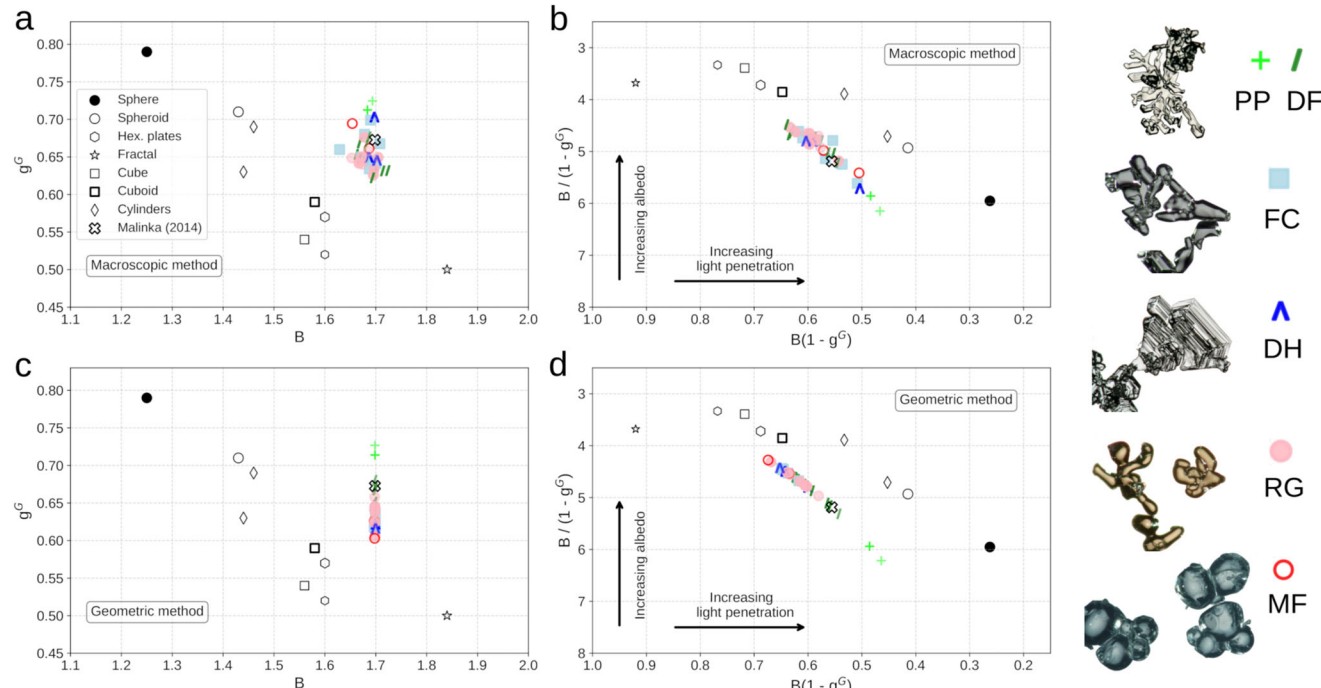

**Fig. 2 | The optical shape of snow. a**, **b** Absorption enhancement parameter B and geometric asymmetry parameter $g^G$ (and combinations) of snow at 900 nm, retrieved with the macroscopic method. **c**, **d** Idem, retrieved with the geometric method. Note that albedo and light penetration depend on other factors than shape, in particular on grain size, so the representation in **b**, **d** must be interpreted at equal snow grain size. In every panel, the dark symbols correspond to geometric shapes reported in the literature (see Supplementary Table 2) and the two-phase random medium, labeled in **a** as Malinka (2014) (see Supplementary Methods 1). The colored ones correspond to the 33 natural snow samples, depending on the snow type[4]: Precipitation Particles (PP), Decomposing and Fragmented precipitation particles (DF), Faceted Crystals (FC), Depth Hoar (DH), Rounded Grains (RG) and Melt Forms (MF).

two-phase random medium, that is $B = 1.70$ at $\lambda = 900$ nm[30]. Over the 600 to 1200 nm range, $B$ barely varies, less than 2% with the geometric method and no more than 7% with the macroscopic method (Supplementary Fig. 2). For wavelengths shorter and longer than 600 nm and 1200 nm, respectively, the macroscopic method is less accurate (see Methods) but the geometric method still yields $B = n^2$. Hence, the absorption enhancement of natural snow do not depend on the microstructural details. Instead, $B$ is virtually constant and equal to $n^2$ over the 400 to 1400 nm range.

The geometric asymmetry parameter $g^G$ of natural snow varies between 0.60 and 0.73 ($0.65 \pm 0.03$) at 900 nm with negligible variations across the visible and NIR range (Fig. 2a, c and Supplementary Fig. 2), while for the geometric shapes it spreads from 0.50 for fractals to 0.79 for spheres. This means that snow is less forward-scattering than spheres, but more than most of the featured geometric shapes. These values agree with those presented in previous studies ($g^G = 0.68$ at $\lambda = 900$ nm[48]; $g^G$ varying from 0.66 to 0.73[40] or 0.64 to 0.66 for a more limited dataset[45]).

The albedo and the light penetration depth are controlled by the grain size (via the specific surface area, SSA) and the combination of $B$ and $g^G$. Indeed, the ratio $B / \text{SSA}(1 - g^G)$ governs the influence of the size and the optical shape on the albedo and the product $B(1 - g^G)\text{SSA}$ on the light penetration[40]. In the representation in Fig. 2b, d, two snowpacks with equal size (i.e. same SSA) but different shapes have the same albedo (respectively light penetration) only if the shapes have the same ordinate (respectively abscissa). In other words, snow samples with equal size but different shape may have different albedo or light penetration. Natural snow spans a region distinct from that of the geometric shapes in the 2D space defined by these quantities (Fig. 2b, d), implying that none of the studied geometric shapes can be used to satisfactorily simulate snow albedo and light penetration at the same time.

Interestingly, a relationship arises between the albedo and the snow type. For a given SSA, fresh snow (PP), like spheres, is a relatively inefficient reflector, while rounded grains (RG) nearly behave as other idealized shapes, such as cylinders (Fig. 2b, d and Supplementary Fig. 3). This is counter-intuitive as rounded grains or melt forms (MF) have the most spherical shape. Regarding light penetration, natural snow behaves similarly to cylinders, and roughly halfway between spheres and fractals (Fig. 2b, d and Supplementary Fig. 3). Fresh snow is, however, more penetrating than the rest of snow types for a given SSA. Even if the spectral albedo of non-spherical shapes can be estimated using spheres by scaling their radius[49], light penetration depth in a medium with spheres is approximately twice longer than in snow with the same SSA. These results show that, in order to represent these quantities, natural snow should not be represented by the geometric shapes that have been commonly implemented in radiative transfer models, and in particular by spheres.

## Towards a universal representation of snow microstructure in optical models

To further understand why all the featured snow types lead to the same constant value for absorption enhancement, we investigate with the ray-tracing model how the value of $B$ varies when an idealized shape is progressively deformed. To this end we explore three shapes: a sphere, a cube and a convex shape without any symmetry, unlike the first two (Supplementary Methods 2 and Supplementary Fig. 4). For the sphere and the convex shape, the number of triangular facets used to generate the surface is gradually decreased, while for the cube, each of the 8 corners is translated in space in a random direction, with an increasing distance. In all cases, this results in increasingly deformed shapes.

The parameters $B$ and $g^G$ of the three shapes at 900 nm, computed with the geometric method, are presented in Fig. 3. Interestingly, for

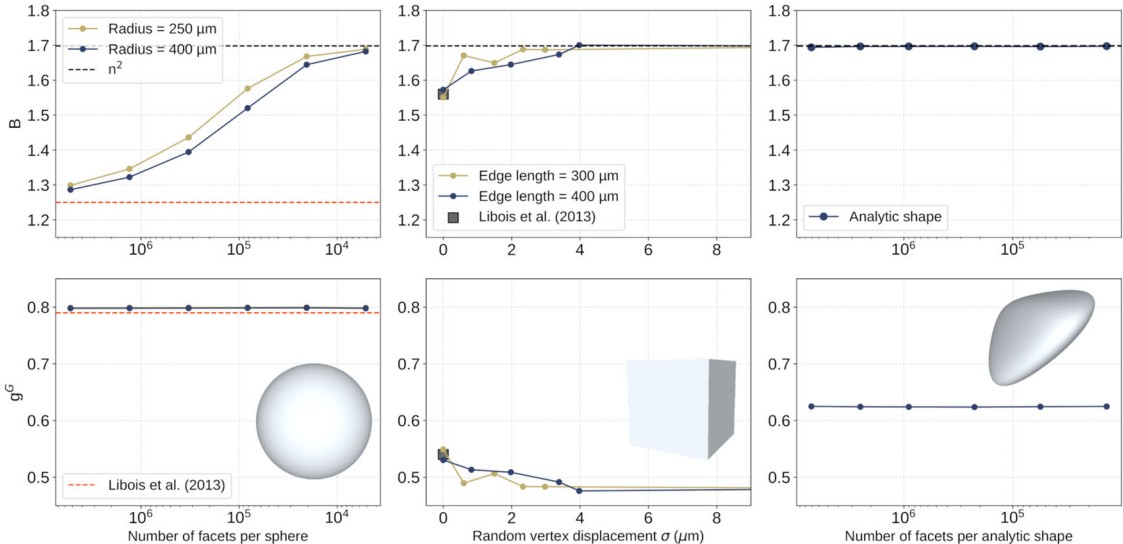

**Fig. 3 | Optical shape parameters of gradually deformed idealized shapes.** Variations of the optical shape parameters (absorption enhancement parameter $B$ (top); geometric asymmetry parameter $g^G$ (bottom)) of a gradually deformed sphere (left), cube (center) and analytical convex shape (right). All the simulations are performed at 900 nm with the geometric method.

the convex shape, $B$ is constant, while for spheres and cubes, $B$ progressively converges as the deformation increases to the value found for natural snow and predicted for the two-phase random medium (1.7 - Fig. 3, top panel). The theoretical value of $B$ for spheres, around 1.25[9,40], is only obtained for the sphere with the largest number of facets (≈5 million), suggesting that even the smallest deviation from this perfection has large consequences for optical properties. This is relevant to understand that, even if rounded grains or melt forms may look spherical, their $B$ and $g^G$ values considerably differ from those of "perfect" spheres.

The underlying reason explaining why and when $B = n^2$ can be established from a series of fundamental studies in mathematics, ecology, optics and nuclear physics[50–53]. The absorption within a weakly-absorbing particle is proportional to the mean path traveled by photons in the particle, and $B$ measures how this distance is increased compared to the propagation in a straight line, in the case of diffuse illumination. $B$ is influenced by two effects, (i) how the photons are focused as they enter the particle (refraction), and (ii) the mean distance traveled by photons in the particle. The first effect introduces a factor $n^2$ and is independent of the particle shape as demonstrated theoretically and experimentally[54,55]. The second effect introduces a factor of exactly 1 (thus leading to $B = n^2$) in several cases: for non-refractive particles ($n = 1$) the photons propagate in straight lines and the mean distance traveled in the particle $<l>$ is given by the Cauchy formula $<l> = 4\,V/S$ ($V$ and $S$ are the volume and surface area of the particle). The same mean distance is obtained for refractive ($n > 1$) particles composed of a scattering material[55] because the Cauchy formula holds for a wide class of random walks[51,53]. The reason is the compensating effect of scattering: longer tortuous paths are balanced by short paths that escape quickly from the particle. However, ice is not a scattering material. In that case, the mathematical theory of billiards can be applied to photons bouncing inside a particle[50], and it was shown[52] that if the photons traverse the entire particle in all directions perfectly uniformly, the mean distance is again given by the Cauchy formula, which implies $B = n^2$. Some billiards (i.e., shapes) are ergodic and verify this isotropy condition for any refractive index. Conversely, idealized shapes such as spheres and cubes, are non-ergodic and some regions may not be uniformly explored by photons coming from the outside, especially if the refractive index is larger than a shape-dependent critical value[52]. As these unexplored regions

generally correspond to very long paths that are only accessible through internal scattering, the mean traveled distance decreases, leading to $B < n^2$ as observed in Fig. 4 for spheres and cubes. Note also that strong absorption also reduces the very long paths, leading to a decreased $B$ (Supplementary Fig. 5). To conclude, the fact that we find $B = n^2$ for all the investigated snow samples in the visible and NIR spectral region strongly suggests that the snow microstructure is fundamentally ergodic.

To investigate whether this result applies to materials other than snow, we computed $B$ for different $n$ values with the ray-tracing model. We find that the value of $B$ for near-perfect spheres closely follows the analytical expression for spheres[9], while the equality between $B$ and $n^2$ stands for less symmetric spheres (Fig. 4a), as well as for the large diversity of snow samples studied here (Fig. 4b). This equality is more general and is actually an experimental evidence that, in terms of absorption enhancement, a weakly absorbing porous material like snow can be represented as a collection of convex particles without symmetry or as a two-phase random medium. Representing such porous media in these ways may be useful and crucial for refining the computation of other shape-dependent optical properties[56], with a wide range of applications well beyond the snow optics community, such as the optical characterization of pharmaceutical powders[57] or solar cell design[58].

In contrast to $B$, $g^G$ is almost unaffected by the shape deformation (Fig. 3, bottom panel), although it is more sensitive to the particle shape or the type of snow (Fig. 2). For spheres, $g^G$ closely matches the value obtained by theoretical calculations[9] and in other studies[27,40]. For cubes, $g^G$ decreases very slightly when the random displacement is applied to the corners, breaking the symmetries between the faces. While $B$ is very sensitive to the rare and long paths discussed above, we conclude that $g^G$ is more sensitive to the few internal reflections experienced by the photons, which determine the ability of snow to scatter light forward. Consequently, $g^G$ does not have a unique value for snow, but mainly spans the interval 0.62–0.68, with a slightly higher value for fresh snow. This range is however much smaller than that estimated from idealized shapes, and importantly, it does not contain the value for spheres (0.79), still commonly implemented in some climate models[36,37]. The values of the two-phase random medium and the convex particle (0.67 and 0.63, respectively) are more consistent but they do not represent the full range of values across the

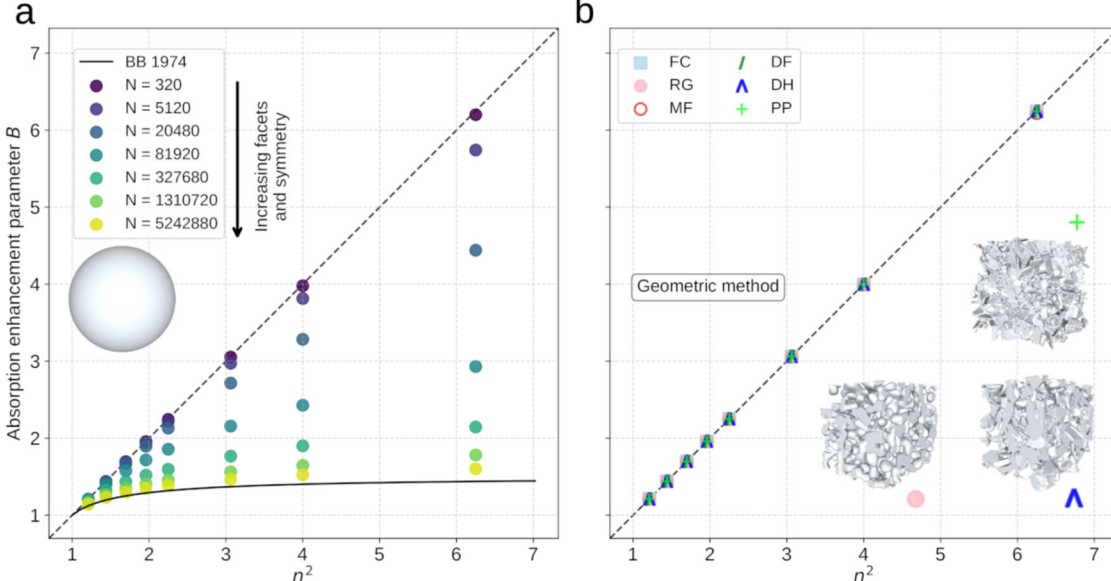

**Fig. 4 | Variations of the absorption enhancement parameter *B* with a varying ice refractive index *n*. a** Variations for spheres. The number of triangular facets used to generate the surface is gradually increased. The BB 1974 curve is the analytical expression for spheres of *B* in terms of *n* (Eq. 9 in[9]). **b** Variations for natural snow. 9 samples are considered, covering most of the main snow types. Three 3D

images of snow samples are shown to illustrate the wide diversity of snow microstructures. The simulations are done with the geometric method, and the ice absorption coefficient γ is kept constant and corresponding to the ice absorption at 900 nm.

diverse types of snow. Even if different *B* and $g^G$ values might be expected for under-represented snow types in our study, such as very peculiar samples of fresh snow or strongly developed depth hoar, our results show that the behavior of snow in the 400 to 1400 nm range is less variable and less shape-dependent than thought so far. This constitutes a step towards a more accurate and universal representation of snow in optical models.

## Discussion

Climate simulations suggest that an ambiguous treatment of snow shape leads to large uncertainties in the estimation of surface radiative forcing[6,7] and global air temperature of up to 1.17 K[8]. This uncertainty is due to the high sensitivity of the global temperature to the snow albedo, enhanced by potent climate feedbacks[59–61]. These uncertainties in snow albedo are driven by *B* and $g^G$ through the single-scattering properties of snow. While these parameters might not be directly used in such relevant models, it is possible to convert these quantities into other optical properties, such as the asymmetry parameter *g* and the single-scattering albedo ω. *g* determines the first order angular variations of the phase function and is directly related to the previously studied $g^G$. The single-scattering co-albedo (1 - ω) depends linearly, for a given particle size, on *B* (see Methods). Only recently snow radiative transfer schemes implemented in climate models, such as SNICAR-ADv3, started to consider non-spherical shapes for *g*, with parameterizations based on spheroids, hexagonal plates and fractals[6,7,38,62]. Still, the spherical assumption is used for *B* (indirectly via ω). This simplification has a direct impact on the albedo, and also strong consequences for the light penetration depth, as spheres are roughly twice more penetrating than natural snow (Fig. 2). This has implications for the thermal regime of the snowpack[63] and for the transmittance of snow over ground[45] or over sea ice[26]. To our knowledge, only in[8] non-spherical values of *B* and $g^G$ are considered. Their 1.17 K global air temperature change was obtained by varying together *g* from 0.89 to 0.78 and *B* from 1.25 to ≈ 1.62 (indirectly via $ω^{64}$).

Here we provide strong observational constraints on *B* and *g* for snow, which shall help reduce uncertainties in climate studies by successfully simulating snow albedo and light penetration at the same

time. *B* is universal and equal to $n^2$, and according to our simulations, the value of *g* for snow over the 33 samples is *g* = 0.82 ± 0.01, with negligible variations in the 400 to 1400 nm range. Although performing climate simulations with these updated values is beyond the scope of this study, we propose a simple estimation of the impact on global air temperature, based on the study from[8] and by considering the quantities represented in Fig. 2b, d (see Methods). By using the constant values of *B* = 1.7 and *g* = 0.82 (or equivalently $g^G$ = 0.65) instead of the values for the non-spherical shape, the simulated global annual-mean air temperature would shift by roughly 0.6 K, assuming similar sensitivity of temperature to snow albedo as in[8]. Moreover, the narrow range of values found here for natural snow would drastically reduce the uncertainties due to the equivocal impact of snow morphology, dropping from 1.17 K to approximately 0.4 K. Beyond climate simulations, the refined knowledge of the optical shape of snow obtained in this study will be beneficial wherever snow optics matters, from snow photochemistry to remote sensing algorithms, solving the long-standing issue of the optical shape of snow.

## Methods

### Snow microstructure from X-ray tomography

The 3D snow microstructure images have been acquired over the last decades by the Centre d'Études de la Neige (Météo-France - CNRS). The dataset presented here (33 images) includes mainly seasonal snow collected in the French Alps, as well as several samples resulting from a series of laboratory metamorphism experiments[32,65–69]. The X-ray tomography was performed at various resolutions, from 4.91 μm to 11.65 μm. Depending on the experimental set-up, the sample volume varied between 2.5 × 2.5 × 2.5 and 9.5 × 9.5 × 9.5 mm³. For image acquisition, samples were first impregnated with liquid 1-chloronaphthalene around −10 °C, and placed below −20 °C, thus forcing the chloronaphthalene to freeze. Small cylinders were then machined at −30 °C, inserted in plexiglas caps and fixed on copper columns for their later insertion into CellStat, a cold cell specifically designed for tomography of frozen samples at ambient temperature[68,70]. After tomography, the differences in X-ray attenuation between the ice, 1-chloronaphthalene and remaining air bubbles

were used to segment the reconstructed 3D grey-level images into 3D binary images. Surface meshes were then obtained through an automated pipeline using iso2mesh[71]. For each image, the function 'v2s' of this mesh generation toolbox was used with the option 'cgalmesh', ensuring the most robust path to product meshes from binary volumes. The maximal deviation from the 0.5 isosurface was set below 0.5 and the maximum radius of the Delaunay spheres was chosen smaller than 15 pixels. For each closed surface, the mesh was then automatically checked for orientation consistency of the facets. A final manual verification and correction step was realized using MeshLab[72].

The whole description of the dataset (type of snow, resolution, etc) is available in the Supplementary Table 1. The corresponding snow microstructural properties (density, SSA) were computed directly over the mesh with the trimesh Python package[73]. Visualizations of several of these images are available in Fig. 1e–g.

## The ray-tracing model

The existing Rough Surface Ray-Tracing (RSRT) model, originally designed to simulate albedo over rough surfaces and topography[46,47], has been adapted to trace light propagation in 3D microstructure images. In this Monte Carlo photon-tracking approach, a set of photons (rays) is followed through the snow from their source to termination (absorption or escape at the top of the sample), relying on the geometric optics approximation (i.e., we consider the microstructure features to be much larger than the wavelength) to simulate the ray path. The light propagation is governed by both absorption within the ice phase (which is wavelength-dependent) and the fundamental laws of reflection and refraction at each air–ice interface, as in[74] and[16]. The origin and initial direction of each ray is randomly generated above the snow microstructure image. When a ray encounters an ice–air interface, the choice between reflection and refraction is random, and depends on the Fresnel's law of reflectance (Supplementary Methods 3). The chosen ray carries all the incident energy. When traveling through the ice phase, part of the energy carried by the ray is lost by absorption, following an exponential decay proportional to the traveled distance in the ice phase (i.e. Beer's law). This is wavelength-dependent as it considers the ice absorption coefficient $\gamma$:

$$\gamma = \frac{4\pi}{\lambda} n_i(\lambda) \qquad (1)$$

where $n_i(\lambda)$ is the imaginary part of the ice refractive index[75].

To simulate a semi-infinite snowpack, a replication algorithm is applied when a ray goes through the boundaries of the original snow microstructure image. Replicating periodically the source image allows simulating a macroscopic snowpack from a single snow microstructure image, largely reducing the computational cost of this approach (in particular with respect to the memory storage of the mesh). It also has some drawbacks, as we consider the snowpack to be homogeneous and single-layer, with snow properties defined by the source microstructure image, such as density or specific surface area (SSA). However, this does not prevent the use of the intrinsic $B$ and $g^G$ quantities in a multi-layered snowpack, where the properties vary vertically[76].

Every single ray is traced until it escapes the simulated semi-infinite snowpack in the upward direction, or until its energy goes below a defined, very low threshold (we consider the ray to be absorbed). $N = 10^6$ rays are launched in each simulation to reach a reasonable accuracy (uncertainty in this Monte-Carlo framework decreases as $1/\sqrt{N}$).

## $B$ and $g^G$ computation: the macroscopic method

Two different methods have been implemented to derive the optical shape parameters, $B$ and $g^G$, from 3D images. The hereinafter called macroscopic method relies on the reflected energy by the snowpack and the vertical profile of energy in the snowpack. Starting from the approximate asymptotic radiative transfer (AART) theory[27], and with the formalism developed in[40], the bi-hemispherical albedo $\alpha$ (hereafter albedo) and the asymptotic flux extinction coefficient $k_e$ are expressed in terms of the optical shape parameters $B$ and $g^G$ by:

$$\alpha \simeq \exp\left(-4\sqrt{\frac{B\gamma V}{3\Sigma(1-g^G)}}\right) \qquad (2)$$

$$k_e \simeq \frac{\rho}{\rho_{ice}}\sqrt{\frac{3B\gamma\Sigma}{V}(1-g^G)} \qquad (3)$$

from where we formulate $B$ and $g^G$ as:

$$B \simeq \frac{-\rho_{ice}k_e(\lambda)\ln(\alpha(\lambda))}{4\rho\gamma(\lambda)} \qquad (4)$$

$$g^G \simeq 1 + \frac{16k_e(\lambda)}{3\rho SSA\ln(\alpha(\lambda))} \qquad (5)$$

using the snow SSA $= 4\Sigma / (V\rho_{ice})$, $\rho$ the snow density and $\rho_{ice}$ the ice density (i.e. 917 kg m$^{-3}$). $\Sigma$ and $V$ are, respectively, the average projected area and the average volume of a particle. Albedo differences of 0.002 are found between a Mie+DISORT model (e.g.[77]) – a robust and accurate radiative transfer model for spherical particles – and the AART theory using $B$ and $g^G$ relevant for spheres (1.25 and 0.79, respectively), and a SSA of 16.4 m² kg$^{-1}$. These albedo differences yield a relative error of $\approx 1\%$ in the $(B, g^G)$ computations ($\approx 0.02$ and $\approx 0.007$, respectively), which we consider negligible. These expressions remain thus valid in the limit of low absorption, which is globally true in the visible and NIR spectral region (400 to 1400 nm).

In the RSRT model we compute the albedo $\alpha$ as the ratio between the number of photons escaping the semi-infinite snowpack with an upward direction and the total number of photons launched. Using a collection of monodisperse spherical particles, the spectral relative error between albedos computed with RSRT and those predicted by the Mie+DISORT model was lower than 3% in the 400 to 1200 nm wavelength range (slightly higher in the 1200 to 1400 nm range - Supplementary Fig. 6). The asymptotic flux extinction coefficient $k_e$ is computed counting the intensity carried by the photons passing through a given horizontal plane z. This intensity shows an exponential decrease with depth, from where we fit a Beer-Lambert law (i.e. $I(z) \simeq I(z=0)\exp(-k_e z)$) to obtain $k_e$ and eventually compute $B$ and $g^G$ with Eqs. 4, 5. The modeling uncertainty of $\alpha$ and $k_e$ is treated here with a Bayesian framework.

## Bayesian treatment of uncertainties

To ensure the accuracy of the computed optical shape parameters with the macroscopic method, we implemented a Bayesian model to quantify the uncertainties. Two optical quantities are derived from the ray-tracing simulations: (i) the albedo, and (ii) the flux extinction coefficient $k_e$. These unknown quantities are then described using some known probability distributions (named priors), which are updated using Bayes' theorem, a process called inference. The resulting posterior distribution provides an estimation of the albedo and $k_e$ modeling uncertainties[78].

Here, for each simulation we describe the prior distribution of albedo with a normal distribution $N(\mu, \sigma^2)$, with $\mu$ being the computed albedo and $\sigma = 1/\sqrt{N}$. For $k_e$, it is less direct. The probability of finding a photon exponentially decreases with depth $z$, so we could describe this as a Bernoulli process. Consequently, the prior distribution of

observing $n$ photons at a certain depth $z$ is given by a Binomial distribution $\mathscr{B}(n, p)$, with $p = \exp(-k_e z)$.

The inference is then performed by means of a Python package[79]. To obtain the posterior estimates, the model fitting is based on samples drawn from the posterior distribution using Markov Chain Monte Carlo (MCMC) methods. In particular, a No-U-Turn Sampler (NUTS) is implemented here (8000 samples). Instead of using only the maximum likelihood estimation, all the posterior estimations of both albedo and $k_e$ are then introduced in Eqs. 4, 5 to obtain a full distribution of the optical shape parameters $B$ and $g^G$ (Supplementary Fig. 1).

## $B$ and $g^G$ computation: the geometric method

We also implemented a more direct approach (called geometric method) where the geometric definition of the shape parameters is computed by means of ray-tracing from the actual snow microstructure. This method relies only on the geometric optics approximation. The absorption enhancement parameter $B$ can be defined by the lengthening of the photon path in the ice phase due to internal reflections and refraction with respect to strictly straight lines[56]. The asymmetry parameter (i.e. $g^G$ in this geometrical optics framework) is defined by the scattering phase function[80,81], and is expressed as:

$$g^G = \frac{1}{2} \int_{-1}^{1} d(\cos \Theta) \cos \Theta p(\Theta) \qquad (6)$$

with $\Theta$ the scattering angle and $p(\Theta)$ the scattering phase function, normalized as:

$$\frac{1}{2} \int_{-1}^{1} d(\cos \Theta) p(\Theta) = 1 \qquad (7)$$

In the RSRT model, in order to compute the scattering phase function, we record the incident and outbound ray direction when entering and going out the ice phase, respectively. The geometric asymmetry parameter $g^G$ is directly deduced from Eq. 6.

## Relating the optical shape parameters to the snow single-scattering properties

The two optical shape parameters, $B$ and $g^G$, can be used in climate modeling. The first step would be to use them to calculate other fundamental snow optical properties, such as the single-scattering albedo $\omega$ and the asymmetry parameter $g$. Defining the single-scattering co-albedo $(1 - \omega)$ as the ratio of absorption to extinction coefficients, and following the formalism developed in[40] (Eqs. (1–6)), we can relate $B$ to $\omega$ by:

$$(1 - \omega) = B\gamma \frac{V}{2\Sigma} \qquad (8)$$

where the snow SSA can be introduced by using $SSA = 4\Sigma / (V\rho_{ice})$:

$$(1 - \omega) = B\gamma \frac{2}{SSA\rho_{ice}} \qquad (9)$$

The asymmetry parameter $g$ is simply the average of the geometric and the diffraction terms ($g^G$ and $g^D$, respectively). In this framework, where the wavelength is small enough compared to the particles, diffraction is mainly forward (i.e. $g^D \approx 1$), so that:

$$g = \frac{g^G + 1}{2} \qquad (10)$$

For more advanced models, $g$ can be used to parameterize the phase function. For instance in DISORT[82], that uses the Legendre polynomial decomposition of the phase function, $g$ appears to be the coefficient of the first order polynomial.

For models that only require the broadband value of the snow albedo, the values of $B$ and $g^G$ found here for natural snow could be used in snow radiative transfer models that rely on such parameters (e.g. TARTES[40]) to eventually derive an updated snow albedo parameterization[83,84].

## Estimation of temperature uncertainty reduction

In[8], it was found that if the shape spans the range from spheres to the Optimized Habit Combination (OHC), the global annual-mean air temperature varies by 1.17 K. Since the snow albedo depends on the ratio $\Gamma = B / (1 - g^G)$, in order to estimate the reduction of the uncertainties related to the optical shape of snow in climate modeling, this quantity and its variations for natural snow are evaluated and compared to those in[8].

Using the values for the sphere ($B = 1.25$, $g^G = 0.79$) and the values for the OHC ($B \approx 1.62$, $g^G \approx 0.56$ -[64]), the $\Gamma$ range explored by[8] is:

$$\Gamma_{R\ddot{a}is\ddot{a}nen} = \frac{B_{sph}}{1 - g^G_{sph}} - \frac{B_{OHC}}{1 - g^G_{OHC}} = 2.27 \qquad (11)$$

In the present study, we obtained a reduced range for natural snow. $B$ can be considered constant and equal to 1.7, and $g^G$ varies mainly between 0.62 and 0.68 ($0.65 \pm 0.03$ - Fig. 2). If we use our values instead of those for the sphere and the OHC, the $\Gamma$ range explored in this study is:

$$\Gamma_{RSRT} = \frac{B_{snow}}{1 - g^G_{snow,upper}} - \frac{B_{snow}}{1 - g^G_{snow,lower}} = 0.84 \qquad (12)$$

From these values we conclude that the shape uncertainty is reduced by a factor $\Gamma_{R\ddot{a}is\ddot{a}nen}/\Gamma_{RSRT} \approx 3$. Assuming linear sensitivity of global temperature to albedo and linear dependency between albedo and $\Gamma$ (valid for small perturbations), the temperature uncertainty is reduced then from 1.17 K to 0.43 K.

## Method limitations
### Snow microstructure.
(1) Our results are based on a finite set of snow images. Although meant to be representative of the diversity of snow, it is limited to 33 images, so that different $B$ and $g$ values cannot be ruled out for very peculiar, under-represented snow morphologies.

(2) Note that only pure snow, i.e. without light-absorbing particles (LAPs), is considered in this study. This is due to the fact that the size distribution of dust particles is much lower than the resolution limit of our X-ray tomography images (a few $\mu m$). A recent study showed the feasibility of capturing the motion of dust particle aggregates in dry snow[85], which could potentially open the way to determine the impact of LAPs on the optical properties of snow by ray-tracing. Work is underway to implement this feature in our ray-tracing model.

### Ray-tracing model.
(1) Wave properties of light (diffraction and polarization) are not considered in our ray-tracing model. However, several studies showed that these geometric optics simplifications do not prevent to correct simulate snow reflectance[16,30,74,86].

(2) Some numerical errors may occur with the most complex snow microstructures (mainly corresponding to fresh snow). This is due to the resolution limit of our current imaging technique, that fails to resolve the most detailed features of such microstructures. This leads to slight artifacts in the mesh generation process and in turn may induce some errors in the photon-tracking method. This is particularly true for the longest photon

paths, at the shortest wavelengths, when the ice absorption is extremely weak.

(3) The definition of a single-scattering event in this study slightly differs from what is typically assumed for unconnected particles[41]. Here, a single-scattering event ends when the ray first exits the ice phase (including reflection at the entrance), whereas in the common definition a ray may enter and exit several times if the particle is concave, before finally escaping the particle. Unfortunately, this common definition requires extracting independent particles from 3D images, which is somewhat arbitrary, since the ice phase is usually mostly connected. Individual snow grains can however be defined as zones separated by regions of potential mechanical weakness (e.g.[87]) though these individual snow grains are still connected by ice. The surface area of these ice-ice contacts is nevertheless small compared to the ice-air interface area[66]. In conclusion, for natural snow, and in order to use the snow microstructure images as is, this uncommon but pragmatic definition of a single-scattering event was used here in the geometric method to derive $g^G$.

### Model uncertainties.

(1) The $B$ and $g^G$ values presented in the text and figures with the macroscopic method correspond to the mean values of the resulting $B$ and $g^G$ distributions. Their dispersion is usually small, with a mean standard deviation over the whole dataset and the 600–1200 nm wavelength range $\overline{\sigma}$ of 0.02 and 0.004 for $B$ and $g^G$ estimations, respectively. For $B$ it is therefore unlikely that the conclusion $B = n^2$ is affected (Supplementary Fig. 1). The same applies to the estimation uncertainty of $g^G$, which is also considerably smaller than the differences between natural snow and the considered geometric shapes.

(2) In relation with the precedent limitation about the modeling uncertainties, the macroscopic method is less accurate at the shortest and longest wavelength range (400–600 nm and 1200–1400 nm, respectively). Below 600 nm, the albedo computation needs to be extremely accurate to derive a precise estimate of the $(B, g^G)$ parameters, which is very computation-costly with a Monte-Carlo approach. This happens because the snow albedo $\alpha$ in this spectral region is close to 1, and in the limit of $\alpha \approx 1$, the $\ln(\alpha)$ dependence becomes very close to zero and in particular, a slight underestimation of the albedo (likely due to the numerical cutoff of the photons) leads to a large overestimation of $B$ (Supplementary Fig. 1). Above 1200 nm, the limitation comes from the underlying asymptotic radiative transfer theory that is only valid in the low absorption limit, which might not be fully respected at these longer wavelengths. However, it is important to note that the geometric method still yields $B = n^2$ below 600 nm and above 1200 nm. For longer wavelengths, the geometric optics approximation (where particles need to have dimensions much larger than the incident wavelength), limits as well the geometric method. To summarize, our results are valid over the 400 to 1400 nm range, which encompasses the most important part of the solar spectrum (≈ 85% of the solar irradiance at the surface is within this range).

(3) In the macroscopic method, some very small uncertainties in the $(B, g^G)$ computation might come from the input macroscopic quantities (density and SSA). In particular, $B$ relies on the snow density $\rho$ (Methods, Eq. 4), and $g^G$ relies on both $\rho$ and SSA (Methods, Eq. 5). These quantities can be estimated with a 2% accuracy[88]. Propagating this uncertainty into the equations, this is equivalent to errors of 0.03 and 0.02 in $B$ and $g^G$, respectively. Moreover, our computed $\rho$ and SSA values compare well to measurements over the voxelized microstructure images in[32], with similar accuracy. However, it is important to note that the geometric method, which does not suffer from these uncertainties in the input macroscopic variables, yields equivalent results to the macroscopic method.

## Data availability

The generated geometric shapes have been deposited in the PerSCIDO platform and are available from https://doi.org/10.18709/perscido.2023.06.ds392. Source data are provided in this paper.

## Code availability

The simulation results and codes to generate the figures have been deposited in the PerSCIDO platform and are available from https://doi.org/10.18709/perscido.2023.06.ds392. The ice refractive index is computed with the snowoptics library available from https://github.com/ghislainp/snowoptics (last access: 16 June 2023) and https://doi.org/10.5281/zenodo.3742138.

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

## Acknowledgements

This research has been supported by the Agence Nationale de la Recherche (MiMESis-3D project—grant no. ANR-19-CE01-0009). IGE and CNRM/CEN are part of Labex OSUG@2020 (Investissements d'Avenir - grant no. ANR-10-LABX-0056). Marie Dumont has received funding from the European Research Council (ERC) under the European Union's Horizon 2020 research and innovation program (IVORI - grant no. 949516). We are also grateful to the 3SR tomographic plateform and the ESRF ID19 beamline where all the 3D images used in this study have been acquired. Most of the computations presented in this paper were performed using the GRICAD infrastructure (https://gricad.univ-grenoble-alpes.fr), which is supported by Grenoble research communities. The authors would like to acknowledge Dr. Kenneth G. Libbrecht, Florent Domine and Météo-France for allowing the use of their photographs in this study.

## Author contributions

A.R., G.P., and M.D. designed the study. G.P. and A.R. developed the ray-tracing model and the estimation method. F.F. conducted the X-ray tomography measurements, prepared the 3D mesh images, and obtained funding. A.R., G.P., M.D., and L.A. performed the analysis of the simulations. Q.L., G.P., A.R., and M.D. explored the relationship $B = n^2$. A.R. drafted the original manuscript, and all authors made suggestions for the improvement of the manuscript.

## Competing interests

The authors declare no competing interests.
