## [Peer Review File · Nature Communications]

REVIEWER COMMENTS

Reviewer #1 (Remarks to the Author):

The importance of snow shape in climate modeling has been often under-appreciated and under-studied in the past, due to limited observations and model incapability of treating complex non-spherical snow grain structures. In recent years, several parameterizations have been developed to represent the impact of non-spherical snow grains on snow radiative transfer and albedo calculations, but still assuming idealized non-spherical shapes. In this submitted manuscript, the authors presented a nice modeling and analysis of snow optical parameters based on observed 3-D images of natural snow at the micrometer scale. They found that previously-assumed idealized snow grain shapes are inaccurate compared to their estimated optical shapes, which more closely approximates a collection of convex particles without symmetry. The results from this study have critical implications for advancing snow albedo modeling and remote sensing retrievals as well as climate modeling through the strong snow albedo feedback. Overall, this is a very nice and important work. Here, I have a few comments and suggestions for the authors to consider.

Major comments:

1. The two key snow optical parameters (B and gG) constrained by observations in this study are typically used in certain snow radiative transfer and albedo calculation algorithm (e.g., asymptotic approximation radiative transfer theory). However, these two optical parameters are not universal parameters used in all snow radiative transfer models/algorithms. Some snow and climate models adopt different snow radiative transfer algorithms (e.g., traditional two-stream, adding-doubling, etc.) which do not directly use these two snow optical parameters. So how to quantitatively convert from B and gG to the snow optical properties (i.e., extinction cross section, single-scattering albedo, asymmetry factor) used by those relevant snow and climate models? This is very important for the future application of this study in climate models.
2. The derivation of B and gG relies on the use of the asymptotic approximation radiative transfer theory, where the theory itself uses some assumptions for simplification in order to obtain the theoretical expressions (e.g., Equations 2 and 3 in the Method). As such, what would be the uncertainty associated with this derivation of B and gG ?
3. The authors mentioned that the RSRT model used as a key component in this study only considers the snowpack to be homogeneous and single-layer, with snow properties defined by the source microstructure image, such as density or specific surface area. How would this introduce additional uncertainty to the model simulations? Or is there any validation/evaluation of this model done by previous studies already to quantify the uncertainty and accuracy?
4. Pages 5-6: "Fresh snow is however more penetrating than the rest of snow types." This is an interesting point. Based on some previous studies (e.g., Liou et al., 2014: <https://doi.org/10.1002/2014JD021665>; He and Flanner, 2020: https://doi.org/10.1007/978-3-030-38696-2_3), it seems that fresh snow (with a more fractal shape) tends to have weaker forward scattering and hence less penetration than its counterpart with other shapes (e.g., rounded shapes), which is the opposite of what has been found here. Is this because of the difference in using idealized 2-D shapes in previous studies and the 3-D structures in this study? Intuitively, fresh snow has a higher albedo and less penetration. How to understand the argument shown above?

Minor comments:

1. Page 1: “reducing by 5 the uncertainties”. The uncertainties of what? Some clarifications are needed here.

2. Page 4, Second paragraph: Any specific reasons for using the 900-nm results (B and gG) here as a typical demonstration/example for detailed analysis?

3. Page 5: “Over the 600 to 1200 nm range, B barely varies ...”. What about at wavelengths outside this 600-1200 nm range over the solar spectrum? Is B still constant? If not sure, then I would suggest adding the wavelength range considered by this study to the abstract for clarification.

4. Page 5: “This is counter-intuitive as rounded grains or melt forms (MF) have the most spherical shape.” This is really interesting. How to understand this counter-intuitive result from a physical perspective?

5. Pages 6-7: A wavelength range (600-1200 nm? 900-nm?) needs to be added when making arguments/conclusions about the B and gG features in these paragraphs.

6. Figures 3 and 4: What are the wavelengths used here?

7. Page 8: “the optical behavior of snow is less variable and less shape-dependent than thought so far”. Is this true for all wavelengths and all real-world snow samples? I would suggest adding a phrase to limit this argument to the situations investigated by this study before directly speculating/generalizing it to all cases/environments.

Reviewer #2 (Remarks to the Author):

GENERAL COMMENTS

This paper combines radiative transfer modelling with an extensive set of 3D observations of snow microstructure to determine the values of two shape-dependent optical parameters of snow, namely the absorption enhancement parameter B and the geometric asymmetry parameter $g_{G\text{sup}}$ at wavelengths between 600 and 1200 nm. When augmented with information about specific surface area (or effective snow grain size) and snow density, these parameters can be used to compute snow albedo and the penetration of solar radiation into snow in climate models and other applications. It is found that the parameters $g_{G\text{sup}}$, and in particular B, vary surprisingly little, and theoretical analysis is provided to support the latter finding. It is also shown that spheres, which are still quite often used for modelling radiative transfer in snow, overestimate $g_{G\text{sup}}$ and underestimate B. While these biases partially compensate each other in the computation of snow albedo, they both act to exaggerate the penetration of solar radiation deeper in the snow. Other idealized shapes also fail to represent either $g_{G\text{sup}}$ or B, or both, accurately, but an assumption of non-symmetric convex particles works rather well.

While many papers have been written on the topic of snow optical

properties, the combination of rigorous modelling and a large set of cutting-edge snow observations makes this study quite compelling. Some suggestions for improving the details are given below.

SPECIFIC COMMENTS

1. p. 2, line 15-16: I don't think "less predictable" is the ideal way of saying this (predicting the shapes/optical properties of ice crystals is also a challenge for models!). Suggestion: "...interaction of sunlight with snow on the ground is more complicated (or difficult) to model than that with ice clouds in the atmosphere".

2. p. 3, 2nd last line: mention already here that RSRT was extended to simulate the radiative transfer within snow (or at least refer to the Methods section). The original RSRT computes only the interaction with the facets at snow surface, assuming that the reflectance from the snow can be modelled using the asymptotic radiative transfer theory (ART).

3. p. 4, line 1: A large part of the solar energy spectrum falls outside the spectral region 600 to 1200 nm considered here. This leads to two questions. (1) What is the reason for limiting the analysis to this spectral range (I suppose that the upper limit is related to the assumptions of the ART)? (2) How should (e.g.) climate modellers apply your findings to compute the snow optical properties over the whole solar spectrum (for this is what they would need to do)? Would the recipe start with computing the single-scattering properties of non-symmetrical convex particles at all relevant wavelengths? I think you should briefly address these questions in the Discussion part.

4. p. 4, 7th last line: please specify the meaning of the uncertainty limits (\pm).

5. p. 5, 6th last line. This sentence should be put in its proper context: "For a given SSA, fresh snow (PP) ...". In reality, fresh snow tends to be highly reflective due to its large SSA.

6. p. 7, last 3 lines: " g is more sensitive to the morphology ... because the ability to scatter light forward is less influenced by the internal path lengths". I cannot follow the logic of this sentence. Please try to clarify it.

7. p. 8, 15th last line: In fact, the value of B corresponding to the nonspherical snow in Räisänen et al. (2017) is larger than the $B \approx 1.50$ quoted here. The absorption enhancement parameter can be seen in Figure 6c of Räisänen et al. (2015) (the blue curves for the "Optimized habit combination OHC). The value is close to or slightly above $B = 1.6$. Note that the symbol ξ is used instead of B .

REFERENCE:

Räisänen, P., Kokhanovsky, A., Guyot, G., Jourdan, O., and Nousiainen, T.: Parameterization of single-scattering properties of snow, *The Cryosphere*, 9, 1277–1301, <https://doi.org/10.5194/tc-9-1277-2015>, 2015.

8. p. 8 (last 7 lines) and p. 9 (first 4 lines). You should be more explicit about how these effects on temperature were estimated. This should be discussed either in Methods or in the Supplementary Material.

9. p. 8, 3 lines from bottom: The actual temperature response might differ between different climate models (e.g. depending on how much the initial "forcing" due to changed snow grain size and albedo is amplified by feedbacks related to changed snow and sea ice cover). It would be healthy to acknowledge this briefly, e.g. "would shift by roughly 0.5 K, assuming similar sensitivity of temperature to snow albedo as in [7]".

10. p. 11, right after Eq. (5). Please define Σ and V .

11. p. 12, right after Eq. (7). "we record the incident and outbound ray direction when entering and going out of the ice phase". This means that a single-scattering event is finished when the ray first exits the ice phase. I agree this is a reasonable definition for snow, as it is not possible to unambiguously define separate snow grains. At the same time, it might be interesting to note that this actually differs from the way single-scattering properties are typically derived for non-spherical particles such as ice crystals. In the typical approach, particles are fired with rays, and the angular distribution of rays exiting the volume is recorded. The difference is that for non-convex particles (consider e.g. the large particle in your Fig. 1h), some rays might pass through the same particle more than once, and this is still regarded as a single-scattering event.

12. A somewhat philosophical follow-up comment is the following:

The most measurable size information for snow is SSA, while for radiative transfer theory (which assumes separate particles), the most relevant size measure would be the specific projected area SPA. These are equal only for convex particles; for non-convex ones, $SSA > SPA$. But is it actually so that for your definition of $g_{G_{sup}}^{sup}$, where the single-scattering event is finished when the ray first exits the ice phase, the difference between SSA and SPA is irrelevant? (This is just my gut feeling. I'm not 100% sure, and I do not even expect you to necessarily comment on this in the paper. But if you are able make a solid statement about this, please go ahead).

13. p. 13: The last sentence is not clear. The purpose is probably to say that the estimation uncertainty of g (or $g_{G_{sup}}^{sup}$) is so small that it does not influence appreciably the differences between natural snow and the considered geometric shapes?

14. Supplementary Fig. 2: Mention also the snow density values.

15. Supplementary Fig. 4: What is the radius of the sphere considered here?

16. Supplementary Table 1: Also mention the origin of the snow samples (geographic region, laboratory-generated etc.)?

17. Supplementary Table 2: Kokhanovsky and Macke (1997) is missing from the reference list of the Supplement.

LANGUAGE CORRECTIONS AND OTHER TECHNICAL ISSUES

1. p. 3, line 18: it should be "simulation results".

2. p. 5, 11 lines from bottom: replace "a distinct region than"

with ``a region distinct from".

3. p. 15, Reference [7]: ``Kirkevåg" should be ``Kirkevåg".

4. Some of the figures and especially the labels are painfully small to read. This applies, at least, to Fig. 2, Fig. 3, and Fig. S5. Please try to enlarge them.

Reviewer #3 (Remarks to the Author):

see attached file

Review for *Nature Communications* by Stephen Warren, March 2023.
Robledano, Picard, Dumont, Flin, Arnaud, Libois: “Unraveling the optical shape of snow.”

General statement:

This is an important paper. It provides evidence that the great variety of crystal shapes in natural snowpacks is not an impediment to accurate prediction of the radiative properties of snowpacks, because all natural shapes have the same “absorption-enhancement parameter” B , and that B is just the square of the refractive index n . The geometric asymmetry factor g^M is more variable, but for visible wavelengths it is still confined within the limits 0.6-0.73. These results were determined by Monte-Carlo simulations applied to three-dimensional microstructure images obtained from X-ray tomography of natural snowpacks.

Only minor revisions are required, but I hope that the authors will undertake an optional expansion of the paper to increase its value, as indicated below in the “major comments”.

Major comments:

(1) How general are these conclusions? The spectral range considered is 600-1200 nm, but ~35% of the solar energy reaching the surface is at shorter wavelengths 300-600 nm, and ice becomes significantly absorptive for solar radiation in the near-IR, 1200-1800 nm. It would therefore be good to expand the presentation to this wider wavelength region. Furthermore, Figure S1 is cut off just where it becomes interesting, both at the short-wavelength end with a sudden sharp rise in B at 600 nm, and at the long-wavelength end with an abrupt drop at 1200 nm.

(2) The reason for the constancy of $B=n^2$ is somewhat mysterious. On pages 6-7, the authors marvel at the fact that “only the smallest deviation from perfection” causes B to increase significantly, but the reader is left wondering why. Can you give more insight?

(3) Figures 2b and 2d show a spread of values along the axis of “increasing albedo”. Since the absorption is nonzero at wavelength 900 nm, the albedo depends on the specific surface area (SSA). But neither the SSA nor the dimensions of the ideal shapes (from which SSA could be computed) are given in Figure 2. For a fair comparison, the ideal shapes (sphere, cylinder, cube, plate) should all have the same SSA. Is that done in Figure 2? Figure 2 shows that $B=1.25$ for the sphere, 1.43 for the cylinder, and 1.6 for the plate. This spread of values seems to conflict with the conclusion of Grenfell & Warren (1999, Figure 4) and Neshyba et al. (2003 Figures 4,5,6) that the single-scattering coalbedo ($1-\bar{\omega}$) is accurate in the equivalent-sphere representation using equal-SSA. If these ideal shapes do not have the same SSA, then Figure 2 should be redrawn for shapes that do.

(4) In Figures 2b and 2d, the sphere is shown to have lower albedo than the other shapes. Dang et al. (2016) showed that for albedo, a model snowpack of spheres can mimic a real snowpack of non-spheres by using a smaller radius r for the sphere: the too-large g can be compensated by reducing r , which increases the SSA. Dang et al. did not examine transmittance, and the arguments in this new paper indicate that the same radius-adjustment would predict too much transmittance, as was pointed out by Libois et al. (2013). Reference 7 (Räisänen et al.) included an extensive discussion of the Dang paper.

But in our prior examination of the equivalent-sphere representation, the transmittances for clouds and snow were not systematically biased high; the errors were small and were both

positive and negative of comparable magnitude, as plotted by Grenfell & Warren (1999, Figure 9), and by Neshyba et al. (2003, Figures 11, 14, 17). So I am puzzled. Any insight you can offer would be appreciated.

(5) Figure 2d shows that fresh snow has the lowest albedo. Commenting on this result on page 5 (six lines from the bottom), the authors say that fresh snow is a “relatively inefficient reflector”. This result is puzzling, since fresh snow probably has a larger SSA than the other snow types that were measured. Some discussion and explanation would be appreciated.

(6) In Figure S4b, the median internal path length for the deformed sphere is larger than the median for the near-perfect sphere, by the factor 1.11. But for these shapes in Figure 3, B differs by a much larger factor, $1.7/1.29=1.32$. Why?

Minor comments:

Introduction, line 2. Many ice particles in clouds do not have “near-perfect geometric shapes”, as shown by the cloud-particle imager (CPI), e.g. Lawson et al. (2011).

Figures 2, 3, and 4 are hard to read because they are too small. For publication, they should be expanded, perhaps rotated so that they can run the full length of a page.

Figure 2. Define the abbreviations PP, DF, FC, DH, RG, MF. And in line 3 of the caption, after “two-phase random medium”, insert “labelled in (a) as Malinka (2014)”.

Page 4, last line. Cite reference 29 here.

Page 5, 5 lines from bottom. “Rounded grains nearly behave as the other idealized shapes”. I think you instead mean that they behave as the other *natural* shapes, not the idealized shapes (sphere, cube, cylinder).

Page 6 lines 3-6. This statement that spheres are useless to represent snow in radiative transfer models is too strong. Dang et al. (2016) showed how spheres can indeed represent nonspherical snow shapes for the purpose of spectral albedo, if the spherical radius is scaled.

Figure 3. It would be helpful to plot also the ratio of facet-size to wavelength. For example, for a sphere of $r=250\ \mu\text{m}$ and 10^5 facets, the radius of a facet is approximately $1.6\ \mu\text{m}$, and the ratio of facet-radius to wavelength is about 1.8.

References:

- Dang, C., Q. Fu, and S.G. Warren, 2016: Effect of snow grain shape on snow albedo. *J. Atmos. Sci.*, 73, 3573-3583. doi:10.1175/JAS-D-15-0276.1
- Lawson, R.P., et al., 2011: Deployment of a Tethered Balloon System for Cloud Microphysics and Radiative Measurements at Ny-Ålesund and South Pole. *J. Atmos. Ocean. Technol.*, 28, 656-670.

Answer to Reviewer #1

Reviewer #1 comment on **NCOMMS-23-08691-T: *Unraveling the optical shape of snow***, by Alvaro Robledano et al., Nature Communications (in review).

We greatly appreciate the following remarks, and we thank the reviewer for their efforts in reviewing our study. We have considered with attention all the comments and suggestions and have attempted to address them as best we can, as detailed below.

The reviewer's comments are reported in black, and our answers are written in blue. The modifications and corrections in the manuscript are reported in green (the unchanged parts of the text are in blue). The page/line numbers, section numbers and figures correspond to those of the original manuscript.

Major comments:

1. The two key snow optical parameters (B and g^G) constrained by observations in this study are typically used in certain snow radiative transfer and albedo calculation algorithm (e.g., asymptotic approximation radiative transfer theory). However, these two optical parameters are not universal parameters used in all snow radiative transfer models/algorithms. Some snow and climate models adopt different snow radiative transfer algorithms (e.g., traditional two-stream, adding-doubling, etc.) which do not directly use these two snow optical parameters. So how to quantitatively convert from B and g^G to the snow optical properties (i.e., extinction cross section, single-scattering albedo, asymmetry factor) used by those relevant snow and climate models? This is very important for the future application of this study in climate models.

We acknowledge that the transferability of these results to other snow radiative transfer models was not sufficiently detailed in the original manuscript. Our results have a general character and apply more or less directly to models that do not explicitly use B and g^G . We agree with the reviewer that B and g^G are not universal, but they are directly or indirectly related to the fundamental single-scattering properties of snow that most radiative transfer (RT) models use, namely the phase function, the scattering and absorption coefficients and/or the single scattering albedo.

The asymmetry parameter g , which is related to the geometric asymmetry parameter g^G computed in our study (c.f. beginning of the Discussion), describes the first order angular variations of the phase function. In the case of a highly diffuse material as snow, this first order is sufficient to compute the albedo and light penetration in the

visible and near infrared (Libois et al., 2013). This is why many RT models are using g directly to describe the phase function. For those models that instead are using the full phase function, it should be possible to make an adjustment to match the g^G obtained in this study.

Using the absorption enhancement parameter B is a bit less simple, but it determines the absorption coefficient and hence the single-scattering albedo ω (Bohren and Nevitt, 1983; Kokhanovsky and Zege, 2004). Considering particles as incoherent scatterers, the parameter B is related to ω by:

$$(1 - \omega) = B \gamma \frac{V}{2\Sigma},$$

where Σ and V are geometrical properties of the particles related to the size (respectively, the average projected area and the average volume of a particle) and γ is the ice absorption coefficient (see Methods, eq. (1) in the original version of our manuscript). B appears as a scaling factor for the “shape” of the particles and the absorption of the constitutive materials. Any RT model considering particles, with known size (via the specific surface area SSA), can use our B values for natural snow to re-parametrize their single scattering albedo or absorption coefficient.

So, while the B and g optical parameters are not universal, almost all radiative transfer models use them at least indirectly, and it is possible to easily convert these quantities into widely-used single-scattering properties of snow. We have expanded the part of the Discussion section where this is now addressed explicitly. We also added a new subsection in the Methods section to provide hints for the application of our results to other radiative transfer models.

The revised discussion of the manuscript now reads:

[...] This uncertainty is due to the high sensitivity of the global temperature to the snow albedo, enhanced by potent climate feedbacks (Qu and Hall, 2007; Flanner et al., 2011; Riihelä et al., 2021). These uncertainties in snow albedo are driven by B and g^G through the single-scattering properties of snow. While these parameters might not be directly used in such relevant models, it is possible to convert these quantities into other optical properties, such as the asymmetry parameter g and the single-scattering albedo ω . g determines the first order angular variations of the phase function and is directly related to the previously studied g^G . The single scattering co-albedo $(1 - \omega)$ depends linearly, for a given particle size, on B (see Methods). Only recently snow radiative transfer schemes implemented in climate models, such as SNICAR-ADv3, started to consider non-spherical shapes for g , [...]

And the new subsection in the Methods section:

Relating the optical shape parameters to the snow single-scattering properties

The two optical shape parameters, B and g^G , can be used in climate modeling. The first step would be to use them to calculate other fundamental snow optical properties, such as the single-scattering albedo ω and the asymmetry parameter g . Defining the single-scattering co-albedo $(1 - \omega)$ as the ratio of absorption to extinction coefficients, and following the formalism developed in Libois et al. (2013) (Eqs. (1) to (6)), we can relate B to ω by:

$$(1 - \omega) = B \gamma \frac{V}{2\Sigma},$$

where the snow SSA can be introduced by using $SSA = 4\Sigma / (V\rho_{ice})$:

$$(1 - \omega) = B \gamma \frac{2}{SSA \rho_{ice}}$$

The asymmetry parameter g is simply the average of the geometric and the diffraction terms (g^G and g^D , respectively). In this framework, where the wavelength is small enough compared to the particles, diffraction is mainly forward (i.e. $g^D \sim 1$), so that:

$$g \simeq \frac{1}{2} (g^G + 1)$$

For more advanced models, g can be used to parametrize the phase function. For instance in DISORT (Stamnes et al., 1988), that uses the Legendre polynomial decomposition of the phase function, g appears to be the coefficient of the first order polynomial.

For models that only require the broadband value of the snow albedo, the values of B and g^G found here for natural snow could be used in snow radiative transfer models that rely on such parameters (e.g. TARTES – Libois et al., 2013) to eventually derive an updated snow albedo parameterization (Gardner and Sharp, 2010; Dang et al., 2015).

References:

Bohren, C. F. and Nevitt, T. J.: Absorption by a sphere: a simple approximation, *Appl. Optics*, 22, 774-775, <https://doi.org/10.1364/AO.22.000774>, 1983.

Dang, C., R. E. Brandt, and S. G. Warren: Parameterizations for narrowband and broadband albedo of pure snow and snow containing mineral dust and black carbon, *J. Geophys. Res. Atmos.*, 120, 5446–5468, <https://doi.org/10.1002/2014JD022646>, 2015.

Gardner, A. S., and M. J. Sharp: A review of snow and ice albedo and the development of a new physically based broadband albedo parameterization, *J. Geophys. Res.*, 115, F01009, <https://doi.org/10.1029/2009JF001444>, 2010.

Kokhanovsky, A. A. and Zege, E. P.: Scattering optics of snow, *Appl. Optics*, 43 (7), 1589–1602, <https://doi.org/10.1364/AO.43.001589>, 2004.

Libois, Q., Picard, G., France, J. L., Arnaud, L., Dumont, M., Carmagnola, C. M., and King, M. D.: Influence of grain shape on light penetration in snow, *The Cryosphere*, 7, 1803–1818, <https://doi.org/10.5194/tc-7-1803-2013>, 2013.

Stamnes, K., Tsay, S. C., Jayaweera, K., and Wiscombe, W.: Numerically stable algorithm for discrete-ordinate-method radiative transfer in multiple scattering and emitting layered media, *Appl. Optics*, 27, 2502–2509, <https://doi.org/10.1364/AO.27.002502>, 1988.

2. The derivation of B and g^G relies on the use of the asymptotic approximation radiative transfer theory, where the theory itself uses some assumptions for simplification in order to obtain the theoretical expressions (e.g., Equations 2 and 3 in the Method). As such, what would be the uncertainty associated with this derivation of B and g^G ?

First of all, the approximate asymptotic radiative transfer (AART) theory is only used for the macroscopic method, and indeed to obtain Equations 2 and 3. On the contrary, the geometric method does not rely on the AART hypotheses but only on the geometric optics theory. The values of B and g^G obtained with the geometric method are thus only impacted by the uncertainty of ray-tracing and the geometric optics assumption.

For the macroscopic method, qualitatively, the main limiting assumption of the AART theory is that the absorption must be small, or equivalently that the medium must be highly diffusive. This applies well to the case of snow in the visible, near-infrared, and for wavelengths up to about 1250-1400 nm (Fig. R1, see below). This covers the most important part of the solar spectrum relevant for albedo calculations (e.g. for climate models), and it covers the spectral region of our study.

Figure R1. Snow spectral albedo computed with the Mie+DISORT model and the AART theory. SSA stands for specific surface area.

Quantitatively, it is difficult to reply very precisely. Here we show a comparison of albedo calculations in diffuse illumination conditions with the Mie+DISORT model (e.g. Carmagnola et al., 2013) – a robust and accurate radiative transfer model for spherical particles – and the AART theory using B and g^G relevant for spheres (1.25 and 0.79, respectively), and a specific surface area (SSA) of $16.4 \text{ m}^2\text{kg}^{-1}$ (equivalent to an effective radius of $\sim 200 \mu\text{m}$). The results show that the albedo difference is 0.002 at 900 nm and remains under 0.005 up to 1200 nm (only slightly higher in the 1200-1400 nm range). Propagating this error at 900 nm (albedo of ~ 0.8) in the equations to calculate B and g^G , and considering all the other parameters as known, yields a relative error of about 1.1% in the (B , g^G) computations (~ 0.02 and ~ 0.007 , respectively). We consider these errors negligible, or at least of the same order of magnitude as the modeling uncertainties associated with the macroscopic method (see Methods, Bayesian treatment of uncertainties). We have added the following to the Methods:

[...] ρ the snow density and ρ_{ice} the ice density (i.e. 917 kg m^{-3}). Σ and V are, respectively, the average projected area and the average volume of a particle.

Albedo differences of 0.002 are found between a Mie+DISORT model (e.g. Carmagnola et al., 2013) – a robust and accurate radiative transfer model for spherical particles – and the AART theory using B and g° relevant for spheres (1.25 and 0.79, respectively), and a SSA of $16.4 \text{ m}^2\text{kg}^{-1}$. These albedo differences yield a relative error of $\sim 1\%$ in the (B , g°) computations (~ 0.02 and ~ 0.007 , respectively), which we consider negligible. These expressions remain thus valid in the limit of low absorption, which is globally true in the visible and NIR spectral region (400 to 1400 nm). [...]

[...] We also implemented a more direct approach (called geometric method) where the geometric definition of the shape parameters is computed by means of ray-tracing from the actual snow microstructure. This method relies only on the geometric optics approximation. [...]

References:

Carmagnola, C. M., Domine, F., Dumont, M., Wright, P., Strellis, B., Bergin, M., Dibb, J., Picard, G., Libois, Q., Arnaud, L., and Morin, S.: Snow spectral albedo at Summit, Greenland: measurements and numerical simulations based on physical and chemical properties of the snowpack, *The Cryosphere*, 7, 1139–1160, <https://doi.org/10.5194/tc-7-1139-2013>, 2013.

Libois, Q., Picard, G., France, J. L., Arnaud, L., Dumont, M., Carmagnola, C. M., and King, M. D.: Influence of grain shape on light penetration in snow, *The Cryosphere*, 7, 1803–1818, <https://doi.org/10.5194/tc-7-1803-2013>, 2013.

3. The authors mentioned that the RSRT model used as a key component in this study only considers the snowpack to be homogeneous and single-layer, with snow properties defined by the source microstructure image, such as density or specific surface area. How would this introduce additional uncertainty to the model simulations? Or is there any validation/evaluation of this model done by previous studies already to quantify the uncertainty and accuracy?

First, regarding the homogeneity: the assumption of homogeneity is not a limitation of RSRT. Most snow RT models indeed assume that snow is a macroscopically homogeneous scattering medium but calculate the intrinsic snow optical properties (scattering and absorption coefficient, phase function, B , g , g° , etc) from the discrete two-phase nature of the materials (e.g. considering ice particles in air, for instance using Mie scattering theory). This “homogenization” is a very standard approach, and our method for calculating the intrinsic quantities B and g° is designed very similarly to this approach. We use a discrete description of the medium and aim to deduce the

intrinsic B and g^G quantities that are relevant for a macroscopic homogeneous medium. However, this does not prevent the use of these quantities in a multi-layered snowpack, where the properties vary vertically (e.g. Toon et al., 1989). A layered snow microstructure could also be used as input of RSRT. We have stated this more clearly in the revised manuscript:

[...] with snow properties defined by the source microstructure image, such as density or specific surface area (SSA). However, this does not prevent the use of the intrinsic B and g^G quantities in a multi-layered snowpack, where the properties vary vertically (Toon et al., 1989). [...]

Second regarding the uncertainties in the input macroscopic quantities: It is true that the source microstructure image determines several snow properties, which may introduce some uncertainties in the (B, g^G) computation when using the **macroscopic** method. In particular, B relies on the snow density ρ estimate (Methods, Eq. 4), and g^G relies on both the snow density ρ and the snow specific surface area (SSA) estimates (Methods, Eq. 5).

According to Hagenmuller et al. (2016), SSA and density can be estimated with a ~2% accuracy. Propagating this uncertainty into the equations, this is equivalent to errors of 0.03 and 0.02 in B and g^G , respectively. However, these quantities are estimated here directly over the mesh by computing the triangulated surface area and the volume of the mesh. The uncertainty in these calculations is difficult to estimate as they are directly done with the Python library *trimesh*. Preliminary comparisons with density/SSA measurements over the voxelized microstructure images in Dumont et al. (2021) yield very similar results to those estimated with *trimesh*, with relative differences of ~1.5 and ~4% for density and SSA, respectively. However, it is important to note that the geometric method, which does not suffer from these uncertainties in the input macroscopic variables, yields equivalent results to the macroscopic method. We have added the following *Method limitation*:

In the macroscopic method, some very small uncertainties in the (B, g^G) computation might come from the input macroscopic quantities (density and SSA). In particular, B relies on the snow density ρ (Methods, Eq. 4), and g^G relies on both ρ and SSA (Methods, Eq. 5). These quantities can be estimated with a 2% accuracy (Hagenmuller et al., 2016). Propagating this uncertainty into the equations, this is equivalent to errors of 0.03 and 0.02 in B and g^G , respectively. Moreover, our computed ρ and SSA values compare well to measurements over the voxelized microstructure images in Dumont et al. (2021), with similar accuracy. However, it is important to note that the geometric method, which does not suffer from these uncertainties in the input macroscopic variables, yields equivalent results to the macroscopic method.

Figure R2: Snow spectral albedo of monodisperse spherical particles computed with the Mie+DISORT model and the RSRT model.

Third regarding the validation/evaluation of the RSRT model: Figure R2 shows a comparison of albedo calculations in diffuse illumination conditions with the Mie+DISORT model and the RSRT model. We used a collection of monodisperse spherical particles with a SSA of 16.4 m²kg⁻¹ (equivalent to an effective radius of ~ 200 μm), as in our previous response. The albedo differences are rather small (relative error lower than 3%) in the 400 - 1200 nm spectral range (slightly higher in the 1200 - 1400 nm range), which corresponds to the spectral validity of this study. A more elaborated answer about the validity of our results across the solar spectrum can be found in our response to the major comments of Reviewers 2 and 3. In the revised manuscript, we have added Figure R2 as the new Supplementary Figure 6, and we have extended the text as follows:

[...] In the RSRT model we compute the albedo α as the ratio between the number of photons escaping the semi-infinite snowpack with an upward direction and the total number of photons launched. Using a collection of monodisperse spherical particles,

the spectral relative error between albedos computed with RSRT and those predicted by the Mie+DISORT model was lower than 3% in the 400 - 1200 nm wavelength range (slightly higher in the 1200 to 1400 nm range – Supplementary Fig. 6). The asymptotic flux extinction coefficient k_e is computed counting [...]

References:

Dumont, M., Flin, F., Malinka, A., Brissaud, O., Hagenmuller, P., Lapalus, P., Lesaffre, B., Dufour, A., Calonne, N., Rolland du Roscoat, S., and Ando, E.: Experimental and model-based investigation of the links between snow bidirectional reflectance and snow microstructure, *The Cryosphere*, 15, 3921–3948, <https://doi.org/10.5194/tc-15-3921-2021>, 2021.

Hagenmuller, P., Matzl, M., Chambon, G., and Schneebeli, M.: Sensitivity of snow density and specific surface area measured by microtomography to different image processing algorithms, *The Cryosphere*, 10, 1039–1054, <https://doi.org/10.5194/tc-10-1039-2016>, 2016.

Toon, O. B., McKay, C. P., Ackerman, T. P., and Santhanam, K.: Rapid calculation of radiative heating rates and photodissociation rates in inhomogeneous multiple scattering atmospheres, *J. Geophys. Res.*, 94, 16287–16301, <https://doi.org/10.1029/JD094iD13p16287>, 1989.

4. Pages 5-6: “Fresh snow is however more penetrating than the rest of snow types.” This is an interesting point. Based on some previous studies (e.g., Liou et al., 2014: <https://doi.org/10.1002/2014JD021665>; He and Flanner, 2020: https://doi.org/10.1007/978-3-030-38696-2_3), it seems that fresh snow (with a more fractal shape) tends to have weaker forward scattering and hence less penetration than its counterpart with other shapes (e.g., rounded shapes), which is the opposite of what has been found here. Is this because of the difference in using idealized 2-D shapes in previous studies and the 3-D structures in this study? Intuitively, fresh snow has a higher albedo and less penetration. How to understand the argument shown above?

Fresh snow has indeed a higher albedo than aged snows in general, because of its much larger specific surface area (SSA) compared to other snow types. However, our sentence is considering only the shape effect, i.e. comparing fresh and aged snow for the same SSA. We have improved the text to clarify this point and make it more explicit to show that we are comparing snow types with the same SSA throughout the paper:

Interestingly, a relationship arises between the albedo and the snow type. For a given SSA, fresh snow (PP), like spheres, is a relatively inefficient reflector, [...]

[...] Regarding light penetration, natural snow behaves similarly to cylinders, and roughly halfway between spheres and fractals (Fig. 2b/d and Supplementary Fig. S2). Fresh snow is however more penetrating than the rest of snow types for a given SSA. [...]

Regarding the difference between fractals and fresh snow, we found that fresh snow is the most penetrating snow type (for a given SSA). We believe that the differences between fractals and fresh snow might come, as the reviewer suggests, from using a particularly idealized shape to mimic the behavior of a complex, porous medium. While we are confident about our results, it should be noted that fresh snow is under-represented in our collection of snow samples, so we cannot exclude a slightly different optical behavior given the wide spread of shapes in fresh snow (we refer to our first point in the Method limitations section). Also, some details of these complex shapes may not be completely resolved because of the resolution limit of our imaging technique.

Minor comments:

1. Page 1: “reducing by 5 the uncertainties”. The uncertainties of what? Some clarifications are needed here.

We meant the uncertainties in global annual-mean air temperature that are **only** related to the choice of a particular shape, e.g., between spheres and the non-spherical shape in Räisänen et al. (2017). With our proposed values for B and g^G for natural snow ($B = 1.7$; $g^G = 0.65 \pm 0.03$), we reduce the possible range of this pair of values, which could potentially involve a X-factor reduction of the uncertainties related to the shape. With the updated values for the non-spherical shape following the comments of reviewer #2, we have modified the abstract as follows:

[...] can be directly used in climate models, reducing by 3 the uncertainties in global air temperature related to the optical shape of snow.

References:

Räisänen, P., Makkonen, R., Kirkevåg, A., and Debernard, J. B.: Effects of snow grain shape on climate simulations: sensitivity tests with the Norwegian Earth System Model, *The Cryosphere*, 11, 2919–2942, <https://doi.org/10.5194/tc-11-2919-2017>, 2017.

2. Page 4, Second paragraph: Any specific reasons for using the 900-nm results (B and g^G) here as a typical demonstration/example for detailed analysis?

We mainly chose the 900-nm results to be in phase with previous studies (e.g. Libois et al., 2013; Krol and Löwe, 2016). It is a compromise to avoid the higher wavelengths where the assumption of low absorption of ice (theoretical limitation of the AART) may not be fully respected, and the lower wavelengths where the numerical computations are heavy due to the very long trajectories of the photons (low absorption of ice) and where the modeling uncertainties are higher. We have thus added the following to the revised manuscript:

[...] g^G can also be expressed in terms of n , in a less trivial way (Supplementary Text 1). We use here the 900 nm results for a comparison with previous studies (Libois et al., 2013; Krol and Löwe, 2016). These results are as well a compromise between the lower wavelengths where the modeling uncertainties are higher (Supplementary Fig. 1) and the higher wavelengths where the assumption of low absorption of ice is less valid than at shorter wavelengths. The absorption enhancement parameter B of natural snow clearly clusters around 1.7 [...]

References:

Libois, Q., Picard, G., France, J. L., Arnaud, L., Dumont, M., Carmagnola, C. M., and King, M. D.: Influence of grain shape on light penetration in snow, *The Cryosphere*, 7, 1803–1818, <https://doi.org/10.5194/tc-7-1803-2013>, 2013.

Krol, Q. and Löwe, H.: Relating optical and microwave grain metrics of snow: the relevance of grain shape, *The Cryosphere*, 10, 2847–2863, <https://doi.org/10.5194/tc-10-2847-2016>, 2016.

3. Page 5: “Over the 600 to 1200 nm range, B barely varies ...”. What about at wavelengths outside this 600-1200 nm range over the solar spectrum? Is B still constant? If not sure, then I would suggest adding the wavelength range considered by this study to the abstract for clarification.

This comment has been raised by the 3 reviewers, and as a major comment by reviewer #2 and #3. To avoid repeating the same answer, we kindly refer the reviewer to our detailed answer to reviewer #2 **Specific comment #3**.

We have accordingly modified the abstract to clarify the wavelength range considered by of this study:

[...] Besides providing a more realistic representation of snow in the visible and near-infrared spectral region (400 to 1400 nm), this breakthrough can be directly used in climate models, [...]

4. Page 5: “This is counter-intuitive as rounded grains or melt forms (MF) have the most spherical shape.” This is really interesting. How to understand this counter-intuitive result from a physical perspective?

We can honestly say that this point has also puzzled the authors. That was one of the reasons that led to the study of the “perfection” of the geometric shapes, i.e. the section entitled “*Towards a universal representation of snow microstructure in optical models*”. The interesting outcome of this part of the study is the peculiar optical behavior of spheres, which must be represented in a quasi-perfect way to find a B value much lower than n^2 . As stated in the manuscript, a possible physical explanation of this is related to the internal paths of the photons in the ice phase, that require a high-degree of symmetry, only present in “perfect” spheres. So, even if rounded grains or melt forms may look spherical, the absence of such symmetries leads to this considerable difference in the values of B and g^G with respect to “perfect” spheres. We have added the following in the text to make it more explicit:

[...] suggesting that even the smallest deviation from this perfection has large consequences for optical properties. This is relevant to understand that, even if rounded grains or melt forms may look spherical, their B and g^G values considerably differ from those of “perfect” spheres. [...]

Note that we have considerably expanded this part of the manuscript (see our response to reviewer #3 comment #2).

5. Pages 6-7: A wavelength range (600-1200 nm? 900-nm?) needs to be added when making arguments/conclusions about the B and g^G features in these paragraphs.

To be clearer, we have added the following at the beginning of the section:

The parameters B and g^G of the three shapes at 900 nm, computed with the geometric method, are presented in Fig. 3. Interestingly, for the convex shape, B is constant, while for spheres and cubes, B progressively converges as the deformation increases to the value found for natural snow and predicted for the two-phase random medium (1.7 -- Fig. 3, top panel). [...]

6. Figures 3 and 4: What are the wavelengths used here?

The wavelength used in both Figures 3 and 4 is 900 nm. It is actually stated in the figure legends (last line).

7. Page 8: “the optical behavior of snow is less variable and less shape-dependent than thought so far”. Is this true for all wavelengths and all real-world snow samples? I would suggest adding a phrase to limit this argument to the situations investigated by this study before directly speculating/generalizing it to all cases/environments.

Our study applies to the visible and NIR spectral region, and is somewhat limited by the set of snow samples used, where some snow types are under-represented, e.g. those in polar snowpacks or the large variety of shapes in fresh snow. Following this suggestion, we have changed the end of the paragraph accordingly:

Even if different B and g^{\ominus} values might be expected for under-represented snow types in our study, such as very peculiar samples of fresh snow or strongly developed depth hoar, our results show that the behavior of snow in the 400 to 1400 nm range is less variable and less shape-dependent than thought so far. This constitutes a step towards a more accurate and universal representation of snow in optical models.

Answer to Reviewer #2

Reviewer #2 comment on **NCOMMS-23-08691-T**: *Unraveling the optical shape of snow*, by Alvaro Robledano et al., Nature Communications (in review).

We thank the reviewer for taking their time to carefully read and review our manuscript, and we appreciate the constructive comments and suggestions that helped us to prepare an improved revised version of the manuscript. All comments have been addressed below.

The reviewer's comments are reported in black, and our answers are written in **blue**. The modifications and corrections in the manuscript are reported in **green** (the unchanged parts of the text are in **blue**). The page/line numbers, section numbers and figures correspond to those of the original manuscript.

Specific comments

1. p. 2, line 15-16: I don't think "less predictable" is the ideal way of saying this (predicting the shapes/optical properties of ice crystals is also a challenge for models!). Suggestion: "...interaction of sunlight with snow on the ground is more complicated (or difficult) to model than that with ice clouds in the atmosphere".

We agree that this was a naive and perhaps misleading statement from our side. The sentence has been updated as suggested.

2. P. 3, 2nd last line: mention already here that RSRT was extended to simulate the radiative transfer within snow (or at least refer to the Methods section). The original RSRT computes only the interaction with the facets at snow surface, assuming that the reflectance from the snow can be modelled using the asymptotic radiative transfer theory (ART).

This is actually written in the Methods section when presenting the ray-tracing model, as well as the reference to the methods section at the end of the paragraph. We agree with the reviewer that it might be useful to already mention it at this stage of the manuscript, so we have modified the sentence as follows:

[...] *The computations are made with the Rough Surface Ray-Tracer (RSRT) model (Larue et al., 2020; Robledano et al., 2022), which has been extended to simulate light propagation in 3D microstructure images (see Methods). For every image, simulations are run in the visible and near-infrared (NIR) spectral range [...]*

3. P. 4, line 1: A large part of the solar energy spectrum falls outside the spectral region 600 to 1200 nm considered here. This leads to two questions. (1) What is the reason for limiting the analysis to this spectral range (I suppose that the upper limit is related to the assumptions of the ART)? (2) How should (e.g.) climate modellers apply your findings to compute the snow optical properties over the whole solar spectrum (for this is what they would need to do)? Would the recipe start with computing the single-scattering properties of nonsymmetrical convex particles at all relevant wavelengths? I think you should briefly address these questions in the Discussion part.

Regarding the first question, this remark has been raised by the 3 reviewers. Our detailed answer here tries to cover the remarks of the 3 reviewers, and is split in two parts (< 600 nm and > 1200 nm)

(1) For the visible spectral region, we acknowledge that a large part of the solar energy reaching the surface is at shorter wavelengths (i.e. 300-600 nm). The spectral absorption coefficient of ice is one of the primary variables controlling the reflectance and transmittance of snow, and is responsible for the main spectral features of snow albedo. In the visible, the snow albedo asymptotically approaches 1, meaning that very little absorption happens in this spectral region, and as well that pure ice becomes almost transparent (Fig. R3, right). Nevertheless, it is interesting to explore the validity of our results in this domain.

Figure R3. (left) Computed flux extinction coefficient k_e for the 33 snow samples. The lower and upper limits of the envelope wrapping the median value represent the 10th and the 90th-percentile of the estimates for each wavelength; (right) Imaginary part of spectral ice refractive index based on the Picard et al. (2016) database. The 600 and 1200 nm vertical lines represent the initially considered range in this study.

However, in terms of ray-tracing, this implies several limitations. As the ice absorption coefficient (Methods, eq. 1) becomes very low, the computing time increases very significantly (several hours for a single simulation) due to the longer

travel of the photons and their many interactions with the ice. In parallel with this, the determination of B and g^G using the macroscopic method becomes less accurate because it relies on the computed albedo α (Methods, eqs. 4 and 5). More specifically in the limit of $\alpha \sim 1$, the $\ln(\alpha)$ dependence on both equations becomes very close to zero and in particular, a slight underestimation of the albedo (likely due to numerical cutoff of the photons) leads to a large overestimation of B . Extremely accurate simulations are therefore required to compute the albedo accurately, with a large increase in computing time.

To investigate the potential to push the applicability of our method in the 400-600 nm range, we have run more accurate simulations for a snow sample to illustrate the (B , g^G) estimation uncertainties in this spectral region (Fig. R4).

Figure R4. (a) B and (b) g^G estimations with the macroscopic method (after Bayesian treatment). The lines in each violin plot correspond to the extrema, the mean, and the 10th and 90th-percentile of the resulting distributions. The snow sample is I23 (rounded grains, see Supplementary Table 1).

As a conclusion, below 600 nm, the estimation uncertainties of both parameters with the **macroscopic method** considerably rises. This large increase in the estimation uncertainties limits the practical validity of our macroscopic method in this spectral region. We believe that performing much longer computations (in weeks) does not provide sufficient understanding gain considering that our second method, the **geometric method**, does not suffer from these weaknesses. In particular for B , it still yields $B = n^2$, which is an important result of our study. Figure R4 has been added as the new Supplementary Figure 1, and at the same time **it will replace the existing Supplementary Figure 5**.

(2) The upper limit of our study (1200 nm) was mainly determined for two reasons. Firstly, our ray-tracing model relies on the geometric optics approximation, whose principles are valid for light-scattering computations involving particles with dimensions much larger than the incident wavelength (Takano and Liou, 1989). Diffraction can therefore be neglected and is taken a posteriori in consideration with g calculated from g^{G} . Secondly, the 1200 nm limit was also determined by the assumptions of the underlying AART theory, which is valid in the limit of low absorption, that is $1 - \omega \ll 1$. Ice is globally weakly absorbing below 1350 nm (Warren, 2008), but the retrieved flux extinction coefficient k_e rises in an abrupt way when reaching the 1200 nm wavelength (Fig. R2, left), meaning that more and more energy is being absorbed by snow and therefore that the limit of low absorption is less valid (see also the albedo comparison between DISORT and the AART theory and between DISORT and RSRT in our answer to reviewer #1, new Supplementary Figure 6). As with the shortest wavelengths, this second limitation only applies to the macroscopic method. The geometric method still yields $B = n^2$ up to 1400 nm, so we have expanded our results to that spectral region. For higher wavelengths, the geometric optics assumption limits as well the geometric method.

To summarize, our results are valid over the range 400-1400 nm, which encompasses the most important part of the solar spectrum (~85% of the solar irradiance at the surface is within this range), even though only the geometric method is accurate for the lowest and highest wavelengths in this range, typically shorter than 600 nm and longer than 1200 nm, respectively. In the revised manuscript, we have updated our Supplementary Figure 1 with this expanded spectral region, we have stated more clearly the spectral validity of the study, and we have added the following *Methods limitation* point:

In relation with the precedent limitation about the modeling uncertainties, the macroscopic method is less accurate at the shortest and longest wavelength range (400 - 600 nm and 1200 - 1400 nm, respectively). Below 600 nm, the albedo computation needs to be extremely accurate to derive a precise estimate of the (B , g^{G}) parameters, which is very computation-costly with a Monte-Carlo approach. This

happens because the snow albedo α in this spectral region is close to 1, and in the limit of $\alpha \sim 1$, the $\ln(\alpha)$ dependence becomes very close to zero and in particular, a slight underestimation of the albedo (likely due to numerical cutoff of the photons) leads to a large overestimation of B (Supplementary Fig. 1). Above 1200 nm, the limitation comes from the underlying asymptotic radiative transfer theory that is only valid in the low absorption limit, which might not be fully respected at these longer wavelengths. However, it is important to note that the geometric method still yields $B = n^2$ below 600 nm and above 1200 nm. For longer wavelengths, the geometric optics approximation (where particles need to have dimensions much larger than the incident wavelength), limits as well the geometric method. To summarize, our results are valid over the 400 to 1400 nm range, which encompasses the most important part of the solar spectrum (~85% of the solar irradiance at the surface is within this range).

Regarding the second question, the recipe for climate modelers would indeed rely on modifying the spectral single scattering properties of snow in radiative transfer models, if such properties are defined in the models (e.g. He et al., 2017). This is detailed in our response to reviewer 1, in their major comment #1. If the snow albedo is defined only by its broadband value, we would recommend to use our values of B and g^G in snow radiative transfer models/algorithms that rely on such parameters (e.g. TARTES – Libois et al., 2013) to eventually derive an updated snow albedo parameterization in climate models (e.g. Gardner and Sharp, 2010; Dang et al., 2015). A whole new Methods section entitled “*Relating the optical shape parameters to the snow single-scattering properties*” has been added, and we have emphasized the transferability of our results to climate modelers in the Discussion:

[...] This uncertainty is due to the high sensitivity of the global temperature to the snow albedo, enhanced by potent climate feedbacks (Qu and Hall, 2007; Flanner et al., 2011; Riihelä et al., 2021). These uncertainties in snow albedo are driven by B and g^G through the single-scattering properties of snow. While these parameters might not be directly used in such relevant models, it is possible to convert these quantities into other optical properties, such as the asymmetry parameter g and the single-scattering albedo ω . g determines the first order angular variations of the phase function and is directly related to the previously studied g^G . The single scattering co-albedo ($1 - \omega$) depends linearly, for a given particle size, on B (see Methods). Only recently snow radiative transfer schemes implemented in climate models, such as SNICAR-ADv3, started to consider non-spherical shapes for g , [...]

References:

Dang, C., R. E. Brandt, and S. G. Warren: Parameterizations for narrowband and broadband albedo of pure snow and snow containing mineral dust and black carbon, *J. Geophys. Res. Atmos.*, 120, 5446–5468, <https://doi.org/10.1002/2014JD022646>, 2015.

Gardner, A. S., and M. J. Sharp: A review of snow and ice albedo and the development of a new physically based broadband albedo parameterization, *J. Geophys. Res.*, 115, F01009, <https://doi.org/10.1029/2009JF001444>, 2010.

He C, Takano Y, Liou KN, Yang P, Li Q, Chen F: Impact of snow grain shape and black carbon-snow internal mixing on snow optical properties: parameterizations for climate models, *J. Clim.*, 30(24), 10019–10036, <https://doi.org/10.1175/JCLI-D-17-0300.1>, 2017

Takano, Y., and K. N. Liou: Solar radiative transfer in cirrus clouds, part I: Single scattering and optical properties of hexagonal ice crystals, *J. Atmos. Sci.*, 46, 3–18, [https://doi.org/10.1175/1520-0469\(1989\)046<0003:SRTICC>2.0.CO;2](https://doi.org/10.1175/1520-0469(1989)046<0003:SRTICC>2.0.CO;2), 1989.

Warren, S. G. and Brandt, R. E.: Optical constants of ice from the ultraviolet to the microwave: a revised compilation, *J. Geophys. Res.*, 113, D14220, <https://doi.org/10.1029/2007JD009744>, 2008.

4. P. 4, 7th last line: please specify the meaning of the uncertainty limits “±”.

The first value shown here represents the **mean** value of the 33 snow samples at 900 nm, while the “±” corresponds to the **standard deviation** of the distribution of values at 900 nm. This applies to the whole section unless otherwise specified. We have added the following sentence in the revised version:

[...] *The absorption enhancement parameter B of natural snow clearly clusters around 1.7 (mean of the 33 samples ± 1 standard deviation: 1.68 ± 0.02 in Fig. 2a, 1.70 ± 0.00 in Fig. 2c), [...]*

5. P. 5, 6th last line. This sentence should be put in its proper context: “For a given SSA, fresh snow (PP)...”. In reality, fresh snow tends to be highly reflective due to its large SSA.

We thank the reviewers #2 and #3 for pointing this out. It is indeed necessary to add such important information either in the figure or in the main text about the results of Figure 2b/d. We have updated the figure legend, and rewritten as follows:

The albedo and the light penetration depth are controlled by the grain size (via the specific surface area, SSA) and the combination of B and g^G . Indeed, the ratio $B / SSA(1 - g^G)$ governs the influence of the size and the optical shape on the albedo and the product $B(1 - g^G)SSA$ on the light penetration (Libois et al., 2013). In the representation in Fig. 2b/d, two snowpacks with equal size (i.e. same SSA) but different shapes have the same albedo (respectively light penetration) only if the

shapes have the same ordinate (respectively abscissa). In other words, snow samples with equal size but different shape might have different albedo or light penetration. Natural snow spans a region distinct from that of the geometric shapes in the 2D space defined by these quantities (Fig. 2b/d), implying that none of the studied geometric shapes can be used to satisfactorily simulate snow albedo and light penetration at the same time.

Interestingly, a relationship arises between the albedo and the snow type. For a given SSA, fresh snow (PP), like spheres, is a relatively inefficient reflector, [...]

[...] Regarding light penetration, natural snow behaves similarly to cylinders, and roughly halfway between spheres and fractals (Fig. 2b/d and Supplementary Fig. S2). Fresh snow is however more penetrating than the rest of snow types for a given SSA. [...]

6. P. 7, last 3 lines: “ g^G is more sensitive to the morphology ... because the ability to scatter light forward is less influenced by the internal path lengths”. I cannot follow the logic of this sentence. Please try to clarify it.

This sentence was indeed confusing. There are two underlying ideas behind the sentence: first, that g^G is more sensitive to the shape than B (i.e. different shapes and in particular different snow types yield different g^G values). Second, regarding the deformed shapes, g^G is considerably less influenced by the symmetries than B . In the revised version of the manuscript, which now includes a longer discussion on the equality $B = n^2$ (see Reviewer #3 comment #2), we have rephrased as follows:

[...] While B is very sensitive to the rare and long paths discussed above, we conclude that g^G is more sensitive to the few internal reflections experienced by the photons, which determine the ability of snow to scatter light forward. Consequently, g^G does not have a unique value for snow[...]

7. P. 8, 15th last line: In fact, the value of B corresponding to the nonspherical snow in Räisänen et al. (2017) is larger than the $B \sim 1.50$ quoted here. The absorption enhancement parameter can be seen in Figure 6c of Räisänen et al. (2015) (the blue curves for the “Optimized habit combination OHC”). The value is close to or slightly above $B = 1.6$. Note that the symbol ξ is used instead of B .

We thank the reviewer for noting this information. The approximate value of 1.50 was inferred from one of the sentences of Räisänen et al. (2017), e.g. “The effect of smaller g is partially counteracted by β being higher (i.e. ω being lower) for the OHC than for spheres in the first four spectral bands, by 12–30% depending on r_e and the band”. So we estimated B values by considering a $\sim 20\%$ higher co-albedo (β) for the

OHC than for spheres. The reference (Räsänen et al., 2015) proposed by the reviewer provides a more correct value, so we have accordingly modified the manuscript:

[...] obtained by varying together g from 0.89 to 0.78 and B from 1.25 to ~ 1.62 (indirectly via the single-scattering albedo – Räsänen et al., 2015). [...]

References:

Libois, Q., Picard, G., France, J. L., Arnaud, L., Dumont, M., Carmagnola, C. M., and King, M. D.: Influence of grain shape on light penetration in snow, *The Cryosphere*, 7, 1803–1818, <https://doi.org/10.5194/tc-7-1803-2013>, 2013.

Räsänen, P., Kokhanovsky, A., Guyot, G., Jourdan, O., and Nousiainen, T.: Parameterization of single-scattering properties of snow, *The Cryosphere*, 9, 1277–1301, <https://doi.org/10.5194/tc-9-1277-2015>, 2015.

Räsänen, P., Makkonen, R., Kirkevåg, A., and Debernard, J. B.: Effects of snow grain shape on climate simulations: sensitivity tests with the Norwegian Earth System Model, *The Cryosphere*, 11, 2919–2942, <https://doi.org/10.5194/tc-11-2919-2017>, 2017.

8. P. 8 (last 7 lines) and p. 9 (first 4 lines). You should be more explicit about how these effects on temperature were estimated. This should be discussed either in Methods or in the Supplementary Material.

Indeed, these effects on temperature are estimated by assuming a similar sensitivity of temperature to snow albedo as in Räsänen et al. (2017).

In Räsänen et al. (2017), they found that if the shape spans the range from spheres to the Optimized Habit Combination (OHC), the global annual-mean air temperature varies by 1.17 K. Using the values for the sphere ($B = 1.25$, $g^G = 0.79$) and the values for the OHC (updated: $B \sim 1.62$, $g^G \sim 0.56$), since the snow albedo depends on the ratio $\Gamma = B / (1 - g^G)$, the Γ range explored by Räsänen et al. (2017), is:

$$\Gamma_{\text{Räsänen}} = \frac{B_{\text{sph}}}{1 - g_{\text{sph}}^G} - \frac{B_{\text{OHC}}}{1 - g_{\text{OHC}}^G} = 2.27$$

In the present study, we obtained a reduced range for natural snow. B can be considered constant and equal to 1.7, and g^G varies mainly between 0.62 and 0.68

(0.65 ± 0.03 - Fig. 2). If we use our values instead of those for the spheres and the OHC, the Γ range explored in this study is therefore:

$$\Gamma_{RSRT} = \frac{B_{snow}}{1 - g_{snow, upper}^G} - \frac{B_{snow}}{1 - g_{snow, lower}^G} = 0.84$$

From these values we conclude that the shape uncertainty is reduced by a factor $\Gamma_{Räisänen} / \Gamma_{RSRT} \sim 3$. Assuming linear sensitivity of global temperature to albedo and linear dependency between albedo and Γ (valid for small perturbations), the temperature uncertainty is reduced then from 1.17 K to 0.43 K. The reviewer will notice that this differs from the 5-factor estimated in the original manuscript, and this is mainly because of the updated B value of the OHC and a small typo in our original estimation.

The following subsection has been added to the Methods section:

Estimation of temperature uncertainty reduction

In Räisänen et al. (2017), it was found that if the shape spans the range from spheres to the Optimized Habit Combination (OHC), the global annual-mean air temperature varies by 1.17 K. Since the snow albedo depends on the ratio $\Gamma = B / (1 - g^G)$, in order to estimate the reduction of the uncertainties related to the optical shape of snow in climate modeling, this quantity and its variations for natural snow are evaluated and compared to those in Räisänen et al. (2017).

Using the values for the sphere ($B = 1.25$, $g^G = 0.79$) and the values for the OHC ($B \sim 1.62$, $g^G \sim 0.56$ - Räisänen et al., 2015), the Γ range explored by Räisänen et al. (2017) is:

$$\Gamma_{Räisänen} = \frac{B_{sph}}{1 - g_{sph}^G} - \frac{B_{OHC}}{1 - g_{OHC}^G} = 2.27$$

In the present study, we obtained a reduced range for natural snow. B can be considered constant and equal to 1.7, and g^G varies mainly between 0.62 and 0.68 (0.65 ± 0.03 - Fig. 2). If we use our values instead of those for the sphere and the OHC, the Γ range explored in this study is:

$$\Gamma_{RSRT} = \frac{B_{snow}}{1 - g_{snow, upper}^G} - \frac{B_{snow}}{1 - g_{snow, lower}^G} = 0.84$$

From these values we conclude that the shape uncertainty is reduced by a factor $\Gamma_{R\ddot{a}is\ddot{a}nen} / \Gamma_{RSRT} \sim 3$. Assuming linear sensitivity of global temperature to albedo and linear dependency between albedo and Γ (valid for small perturbations), the temperature uncertainty is reduced then from 1.17 K to 0.43 K.

9. P. 8, 3 lines from bottom: The actual temperature response might differ between different climate models (e.g. depending on how much the initial “forcing” due to changed snow grain size and albedo is amplified by feedbacks related to changed snow and sea ice cover). It would be healthy to acknowledge this briefly, e.g. “would shift by roughly 0.5 K, assuming similar sensitivity of temperature to snow albedo as in [7]”.

We agree with this comment and we have accordingly updated the manuscript as suggested by the reviewer. The following also takes into account the previous comment:

[...] we propose a simple estimation of the impact on global air temperature, based on the study from R\ddot{a}is\ddot{a}nen et al. (2017) and by considering the quantities represented in Fig. 2b/d (see Methods). By using the constant values of $B = 1.7$ and $g = 0.82$ (or equivalently $g^G = 0.65$) instead of the values for the non-spherical shape, the simulated global annual-mean air temperature would shift by roughly 0.6 K, assuming similar sensitivity of temperature to snow albedo as in R\ddot{a}is\ddot{a}nen et al. (2017). Moreover, the narrow range of values found here for natural snow would drastically reduce the uncertainties due to the equivocal impact of snow morphology, dropping from 1.17 K to approximately 0.4 K. [...]

10. P. 11, right after Eq. (5). Please define Σ and V .

We have updated the manuscript as follows:

[...] ρ the snow density and ρ_{ice} the ice density (i.e. 917 kg m^{-3}). Σ and V are, respectively, the average projected area and the average volume of a particle. These expressions remain valid in the limit of low absorption [...]

11. P. 12, right after Eq. (7). “we record the incident and outbound ray direction when entering and going out of the ice phase”. This means that a single-scattering event is

finished when the ray first exits the ice phase. I agree that this is a reasonable definition for snow, as it is not possible to unambiguously define separate snow grains. At the same time, it might be interesting to note that this actually differs from the way single-scattering properties are typically derived for non-spherical particles such as ice crystals. In the typical approach, particles are fired with photons, and the angular distribution of photons exiting the volume is recorded. The difference is that for non-convex particles (consider e.g. the large particle in your Fig. 1h), some rays might pass through the same particle more than once, and this is still regarded as a single-scattering event.

Thank you for pointing out this subtle but important difference of definition. We agree that there is a difference in the single-scattering event traditionally defined for single independent particles versus our definition. As the reviewer has pointed out, our definition is uncommon but pragmatic. The priority in this study was to use the microtomographic images as is, in order to get rid of the granular assumption often overused for snow. Also from a practical point of view defining a snow grain and detecting independent grains in microtomographic images is a difficult and ambiguous task. The main benefits of combining 3D observations of snow microstructure with ray-tracing make it difficult if not impossible to use the common single-scattering event of independent particles since the particles are not independent.

Choosing the right definition of a single-scattering event in natural snow (or any other porous medium) is quite challenging (e.g. Tancrez and Taine, 2004; Xiong et al., 2015). We thus chose this pragmatic definition (used for the geometric method only) because it seemed the most fair and reasonable definition of a single-scattering event in natural snow. Speculating on the impact, we think that concave shapes would have a slightly more isotropic scattering with the independent particle definition than with ours (ie. lower g°), and a longer path in the particle (ie. higher B).

We agree with the reviewer that this difference should be stated in a clearer way, so we propose to add the following in the Methods section (in particular, in the Methods limitations sub-section):

The definition of a single-scattering event in this study slightly differs from what is typically assumed for unconnected particles (Saito and Yang, 2019). Here, a single-scattering event ends when the ray first exits the ice phase (including reflection at the entrance), whereas in the common definition a ray may enter and exit several times if the particle is concave, before finally escaping the particle. Unfortunately, this common definition requires extracting independent particles from 3D images, which is somewhat arbitrary, since the ice phase is usually mostly connected. Individual snow grains can however be defined as zones separated by regions of potential mechanical weakness (e.g. Hagenmuller et al., 2014) though these individual snow grains are still connected by ice. The surface area of these

ice/ice contacts is nevertheless small compared to the ice/air interface area (Flin et al., 2011). In conclusion, for natural snow, and in order to use the snow microstructure images as is, this uncommon but pragmatic definition of a single-scattering event was used here in the geometric method to derive g^G .

References:

Flin, F., B. Lesaffre, A. Dufour, L. Gillibert, A. Hasan, S. Rolland du Roscoat, S. Cabanes and P. Pugliese, On the computations of specific surface area and specific grain contact area from snow 3D images, Furukawa, Y., ed., Physics and Chemistry of Ice, Hokkaido University Press, Sapporo, Japan, 321-328, 2011.

Hagenmuller, P., G. Chambon, F. Flin, S. Morin and M. Naaim: Snow as a granular material : assessment of a new grain segmentation algorithm, *Granul. Matter*, 16 (4), 421-432, <https://doi.org/10.1007/s10035-014-0503-7>, 2014.

Saito, M. & Yang, P.: Oriented Ice Crystals: A Single-Scattering Property Database for Applications to Lidar and Optical Phenomenon Simulations, *J. Atmos. Sci.*, 76 (9), 2635–2652, <https://doi.org/10.1175/JAS-D-19-0031.1>, 2019.

Tancrez, M., and Taine, J.: Direct identification of absorption and scattering coefficients and phase function of a porous medium by a Monte Carlo technique, *International Journal of Heat and Mass Transfer*, 47(2), 373-383, [https://doi.org/10.1016/S0017-9310\(03\)00146-7](https://doi.org/10.1016/S0017-9310(03)00146-7), 2004.

Xiong, C., Shi, J., Ji, D., Wang, T., Xu, Y. and Zhao, T.: A New Hybrid Snow Light Scattering Model Based on Geometric Optics Theory and Vector Radiative Transfer Theory, *IEEE Transactions on Geoscience and Remote Sensing*, 53(9), 4862-4875, <https://doi.org/10.1109/TGRS.2015.2411592>, 2015.

12. A somewhat philosophical follow-up comment is the following: The most measurable size information for snow is SSA, while for radiative transfer theory (which assumes separate particles), the most relevant size measure would be the specific projected area SPA. These are equal only for convex particles; for non-convex ones, $SSA > SPA$. But is it actually so that for your definition of g^G , where the single-scattering event is finished when the ray first exits the ice phase, the difference between SSA and SPA is irrelevant? (This is just my gut feeling. I'm not 100% sure, and I do not even expect you to necessarily comment on this in the paper. But if you are able to make a solid statement about this, please go ahead).

The specific projected area (SPA) might be of high interest for radiative transfer theory, that assumes independent and unconnected particles. For such particles (e.g. ice crystals in the atmosphere), it is possible to calculate the SPA whether the particles are convex or non-convex. However, this is less straightforward for snow, if we consider it as a continuous two-phase medium. Finding a rigorous definition of

the SPA for a bicontinuous medium is quite challenging, in contrast to SSA, that is quite straightforward to obtain on 3D microstructure images.

For convex particles, the mean chord length $\langle l \rangle$ is given by $\langle l \rangle = 4V/S = V/P$ (V , S and P are the volume, the surface area and the projected area of the particle; Malinka (2014), Eq. 6). In other words, $SSA = 4*SPA$ (Grenfell et al., 2005). Interestingly, for some non-convex particles (e.g. hollow spheres), it has been shown that $\langle l \rangle = 4V/S$ remains valid (Mazzolo, 2003).

At this point, exploring this might be out of the scope of the study, but we are greatly thankful to the reviewer. We believe that it is an excellent lead to follow in the future to better understand the differences between seeing snow as a granular or as a bicontinuous material, as it actually is. This would in any case require a rigorous definition of SPA for bicontinuous media.

References:

Grenfell, T. C., Neshyba, S. P., and Warren, S. G.: Representation of a nonspherical ice particle by a collection of independent spheres for scattering and absorption of radiation: 3. Hollow columns and plates, *J. Geophys. Res.*, 110, D17203, <https://doi.org/10.1029/2005JD005811>, 2005.

Malinka, A. V.: Light scattering in porous materials: Geometrical optics and stereological approach, *J. Quant. Spectrosc. Ra.*, 141, 14–23, <https://doi.org/10.1016/j.jqsrt.2014.02.022>, 2014.

Mazzolo, A., Roesslinger, B., and Diop, C. M.: On the properties of the chord length distribution, from integral geometry to reactor physics, *Annals of Nuclear Energy*, 30(14), 1391-1400, [https://doi.org/10.1016/S0306-4549\(03\)00084-7](https://doi.org/10.1016/S0306-4549(03)00084-7), 2003.

13. P. 13: The last sentence is not clear. The purpose is probably to say that the estimation uncertainty of G (or gG) is so small that it does not influence appreciably the differences between natural snow and the considered geometric shapes?

Thanks for noting that it was not clear enough. We have rephrased as this:

The same applies to the estimation uncertainty of g^c , which is also considerably smaller than the differences between natural snow and the considered geometric shapes.

14. Supplementary Fig. 2: Mention also the snow density values.

In the updated figure, we have added the density values alongside the snow types and the SSA values.

15. Supplementary Fig. 4: What is the radius of the sphere considered here?

Thanks for pointing this out, as it was missing. The radius of the sphere is 250 μm .

16. Supplementary Table 1: Also mention the origin of the snow samples (geographic region, laboratory-generated etc.)?

The snow samples were either collected in the field (French Alps) or come from controlled cold-room experiments. The “IPx” series was sampled at increasing depths in the snowpack of the Girose glacier (Écrins, French Alps), and the *01iso* sample was collected at Col de Porte (Chartreuse, French Alps). The rest of the samples come from controlled cold-room experiments. In such cases, the initial sample was most of the time recent alpine snow (e.g. Flin et al., 2004; Dumont et al., 2021). We have added the corresponding column in the table, as well as the following in the caption:

The snow samples were either collected in the field (French Alps) or come from controlled cold-room experiments. In the latter case, the initial sample was most of the time recent alpine snow. More details on the snow sampling and characterization are available at each of the correspondent studies.

References:

Flin, F., Brzoska, J.-B., Lesaffre, B., Coléou, C. & Pieritz, R. A.: Three-dimensional geometric measurements of snow microstructural evolution under isothermal conditions. *Annals of Glaciology* 38 (1), 39–44, <https://doi.org/10.3189/172756404781814942>, 2004.

Dumont, M., Flin, F., Malinka, A., Brissaud, O., Hagenmuller, P., Lapalus, P., Lesaffre, B., Dufour, A., Calonne, N., Rolland du Roscoat, S., and Ando, E.: Experimental and model-based investigation of the links between snow bidirectional reflectance and snow microstructure, *The Cryosphere*, 15, 3921–3948, <https://doi.org/10.5194/tc-15-3921-2021>, 2021.

17. Supplementary Table 2: Kokhanovsky and Macke (1997) is missing from the reference list of the Supplement.

Thank you, this has been corrected.

Language corrections and other technical issues

1. P. 3, line 18: it should be “simulation results”.

Thanks, the typo has been corrected.

2. P. 5, 11 lines from bottom: replace a “distinct region than” with “a region distinct from”.

Idem.

3. P. 15, Reference (7): “Kirkevåg” should be “Kirkevåg”.

We are sorry about that. The author’s name has been corrected.

4. Some of the figures and especially the labels are painfully small to read. This applies, at least, to Fig. 2, Fig. 3, and Fig. S5. Please try to enlarge them.

Thanks for noting this. We have slightly modified and enlarged them in the revised version, and if the manuscript is eventually accepted for publication, we will work with the editorial and production team to upgrade them to their standards.

Answer to Reviewer #3 (Stephen Warren)

Reviewer #3 (Stephen Warren) comment on **NCOMMS-23-08691-T**: *Unraveling the optical shape of snow*, by Alvaro Robledano et al., Nature Communications (in review).

We thank the reviewer for taking his time to carefully read and review our manuscript. We appreciate his constructive remarks and comments that helped us to prepare an improved, expanded and revised version of the manuscript. All comments have been addressed below.

The reviewer's comments are reported in black, and our answers are written in blue. The modifications and corrections in the manuscript are reported in green (the unchanged parts of the text are in blue). The page/line numbers, section numbers and figures correspond to those of the original manuscript.

Major comments:

(1) How general are these conclusions? The spectral range considered is 600-1200 nm, but ~35% of the solar energy reaching the surface is at shorter wavelengths 300-600 nm, and ice becomes significantly absorptive for solar radiation in the near-IR, 1200-1800 nm. It would therefore be good to expand the presentation to this wider wavelength region. Furthermore, Figure S1 is cut off just where it becomes interesting, both at the short-wavelength end with a sudden sharp rise in B at 600 nm, and at the long-wavelength end with an abrupt drop at 1200 nm.

This remark has been raised by the 3 reviewers and we have thoroughly worked on it. To avoid repeating the same answer, we refer the reviewer to our detailed answer to reviewer #2 Specific comment #3.

In order to quickly answer this question, our macroscopic method is less valid at the shortest and longest wavelength range (300 - 600 nm, and above 1200 nm, respectively). Below 600 nm, the albedo computation needs to be extremely accurate to derive a precise estimate of the (B, g°) parameters, which is very computationally-costly with a Monte-Carlo approach. This happens because the snow albedo α in this spectral region is close to 1, and in the limit of $\alpha \sim 1$, the $\ln(\alpha)$ dependence becomes very close to zero and in particular, a slight underestimation of the albedo (likely due to numerical cutoff of the photons) leads to a large overestimation of B , which we suspect is the main problem. The initially considered upper limit (1200 nm) was determined by both the underlying AART theory (only valid in the low absorption limit) and the geometric optics framework, where particles need to have dimensions much larger than the incident wavelength. As the

geometric method only suffers from the geometric optics limitations, we have expanded our results to the 400 - 1400 nm spectral region.

In the revised manuscript, besides stating clearly the range of validity of the study throughout the manuscript, we have added the following *Methods limitation* point:

In relation with the precedent limitation about the modeling uncertainties, the macroscopic method is less accurate at the shortest and longest wavelength range (400 - 600 nm and 1200 - 1400 nm, respectively). Below 600 nm, the albedo computation needs to be extremely accurate to derive a precise estimate of the (B , g°) parameters, which is very computation-costly with a Monte-Carlo approach. This happens because the snow albedo α in this spectral region is close to 1, and in the limit of $\alpha \sim 1$, the $\ln(\alpha)$ dependence becomes very close to zero and in particular, a slight underestimation of the albedo (likely due to numerical cutoff of the photons) leads to a large overestimation of B (Supplementary Fig. 1). Above 1200 nm, the limitation comes from the underlying asymptotic radiative transfer theory that is only valid in the low absorption limit, which might not be fully respected at these longer wavelengths. However, it is important to note that the geometric method still yields $B = n^2$ below 600 nm and above 1200 nm. For longer wavelengths, the geometric optics approximation (where particles need to have dimensions much larger than the incident wavelength), limits as well the geometric method. To summarize, our results are valid over the 400 to 1400 nm range, which encompasses the most important part of the solar spectrum (~85% of the solar irradiance at the surface is within this range).

(2) The reason for the constancy $B=n^2$ is somewhat mysterious. On pages 6-7, the authors marvel at the fact that “only the smallest deviation from perfection” causes B to increase significantly, but the reader is left wondering why. Can you give more insight?

It was a decision not to explicitly detail this point, first to keep the manuscript accessible, and second because the authors could not find a way to explain it in a brief but rigorous way.

The conclusions that are drawn in that section of the manuscript are inspired by theoretical works that are not strictly related to snow optics. The underlying reason explaining why and when $B=n^2$ can be established from a series of fundamental studies from different scientific fields. We have worked on explaining this equality and we have thus rewritten as follows:

[...] The theoretical value of B for spheres, around 1.25 (Bohren and Barkstrom, 1974; Libois et al., 2013), is only obtained for the sphere with the largest number of facets (~ 5 million), suggesting that even the smallest deviation from this perfection

has large consequences for optical properties. This is relevant to understand that, even if rounded grains or melt forms may look spherical, their B and g^G values considerably differ from those of “perfect” spheres.

The underlying reason explaining why and when $B = n^2$ can be established from a series of fundamental studies in mathematics, ecology, optics and nuclear physics (Szász, 2017; Blanco and Fournier, 2003; Majic et al., 2021; Reuss, 2018). The absorption within a weakly-absorbing particle is proportional to the mean path traveled by photons in the particle, and B measures how this distance is increased compared to the propagation in a straight line, in the case of diffuse illumination. B is influenced by two effects, (i) how the photons are focused as they enter the particle (refraction), and (ii) the mean distance traveled by photons in the particle. The first effect introduces a factor n^2 and is independent of the particle shape as demonstrated theoretically and experimentally (Yablonovitch, 1982; Savo et al., 2017). The second effect introduces a factor of exactly 1 (thus leading to $B = n^2$) in several cases: for non-refractive particles ($n = 1$) the photons propagate in straight lines and the mean distance traveled in the particle $\langle l \rangle$ is given by the Cauchy formula $\langle l \rangle = 4V/S$ (V and S are the volume and surface area of the particle). The same mean distance is obtained for refractive ($n > 1$) particles composed of a scattering material (Savo et al., 2017) because the Cauchy formula holds for a wide class of random walks (Blanco and Fournier, 2003; Reuss, 2018). The reason is the compensating effect of scattering: longer tortuous paths are balanced by short paths that escape quickly from the particle. However, ice is not a scattering material. In that case, the mathematical theory of billiards can be applied to photons bouncing inside a particle (Szász, 2017), and it was shown (Majic et al., 2021) that if the photons traverse the entire particle in all directions perfectly uniformly, the mean distance is again given by the Cauchy formula, which implies $B = n^2$. Some billiards (i.e. shapes) are ergodic and verify this isotropy condition for any refractive index. Conversely, idealized shapes such as spheres and cubes, are non-ergodic and some regions may not be uniformly explored by photons coming from the outside, especially if the refractive index is larger than a shape-dependent critical value (Majic et al., 2021). As these unexplored regions generally correspond to very long paths that are only accessible through internal scattering, the mean traveled distance decreases, leading to $B < n^2$ as observed in Fig. 4 for spheres and cubes. Note also that strong absorption also reduces the very long paths, leading to a decreased B (Supplementary Fig. 5). To conclude, the fact that we find $B = n^2$ for all the investigated snow samples in the visible and NIR spectral region strongly suggests that the snow microstructure is fundamentally ergodic.

To investigate whether this result applies to materials other than snow, we computed B for different n values with the ray-tracing model. [...]

New Supplementary Fig. 5: (left) Variations of the shape parameter B of a gradually deformed sphere when ice absorption is increased. (right) Ice absorption coefficient in the NIR spectral region. The crosses correspond to the wavelengths shown in the left panel.

References:

Blanco, S. and Fournier, R.: An invariance property of diffusive random walks, *Europhys. Lett.* 61, 168, <https://doi.org/10.1209/epl/i2003-00208-x>, 2003.

Majic, M., Somerville, W. R. C. & Le Ru, E. C. Mean path length inside nonscattering refractive objects, *Phys. Rev. A*, 103, L031502, <https://doi.org/10.1103/PhysRevA.103.L031502>, 2021.

Reuss, P.: Cauchy's theorem and generalization, *EPJ Nuclear Sci. Technol.*, 4, 50, <https://doi.org/10.1051/epjn/2018010>, 2018.

Savo, R., Pierrat, R., Najar, U., Carminati, R., Rotter, S., and Gigan, S.: Observation of mean path length invariance in light-scattering media. *Science*, 358 (6364), 765–768, <https://doi.org/10.1126/science.aan4054>, 2017.

Szász, D.: Multidimensional hyperbolic billiards, Preprint at <https://arxiv.org/abs/1701.02955>, 2017.

Yablonovitch, E.: Statistical ray optics, *J. Opt. Soc. Am.*, 72, 899-907, <https://doi.org/10.1364/JOSA.72.000899>, 1982.

(3) Figures 2b and 2d show a spread of values along the axis of “increasing albedo”. Since the absorption is nonzero at wavelength 900 nm, the albedo depends on the specific surface area (SSA). But neither the SSA nor the dimension of the ideal shapes (from which SSA could be computed) are given in Figure 2. For a fair comparison, the ideal shapes (sphere, cylinder, cube, plate) should all have the same SSA. Is that done in Figure 2? Figure 2 shows that $B = 1.25$ for the sphere, 1.43 for the cylinder and 1.6 for the plate. This spread of values seems to conflict with the conclusion of Grenfell & Warren (1999, Figure 4) and Neshyba et al. (2003 Figures 4,5,6) that the single-scattering coalbedo ($1 - \omega$) is accurate in the equivalent-sphere representation using equal-SSA. If these ideal shapes do not have the same SSA, then Figure 2 should be redrawn for shapes that do.

We recognize that the statement “increasing albedo/light penetration” in Figure 2b/d was misleading and we have solved this issue. A similar comment was raised by reviewer #2, please also see our detailed answer to this point (specific comment #5 in our response to reviewer #2). In the revised manuscript, it is clearly stated that in order to compare different snow types (and shapes), the results are for equal size/SSA. We have updated the figure legend and rewritten this part as follows:

The albedo and the light penetration depth are controlled by the grain size (via the specific surface area, SSA) and the combination of B and g^G . Indeed, the ratio $B / SSA(1 - g^G)$ governs the influence of the size and the optical shape on the albedo and the product $B(1 - g^G)SSA$ on the light penetration (Libois et al., 2013). In the representation in Fig. 2b/d, two snowpacks with equal size (i.e. same SSA) but different shapes have the same albedo (respectively light penetration) only if the shapes have the same ordinate (respectively abscissa). In other words, snow samples with equal size but different shape might have different albedo or light penetration.

We appreciate the reviewer's suggestion to redraw Figure 2 for shapes that have the same SSA. However, for a given shape, the SSA has almost no influence on the value of B and g^G , as described in Libois et al. (2013 - Section 3 p.1807): “All numerical calculations were performed on crystals smaller than 1mm so that absorption within a single grain is very low and the calculated values of B and g^G do not depend on grain size.” This is also shown in our Results section entitled “Towards a universal representation of snow microstructure in optical models” (page 6 and onwards). In Figure 3, spheres and cubes of different sizes yield very similar pairs of (B , g^G) values for each shape. In particular, two almost-perfect spheres of ~ 13 and $\sim 8 \text{ m}^2\text{kg}^{-1}$ (radius of 250 and 400 μm , respectively) both yield $B \sim 1.29$ and $g^G \sim 0.80$, while two perfect cubes of SSA ~ 22 and $\sim 16 \text{ m}^2\text{kg}^{-1}$ (edge length of 300 and 400 μm , respectively) have $B \sim 1.56$ and $g^G \sim 0.54$.

The papers by Grenfell & Warren (1999) and Neshyba et al. (2003) demonstrate that the S/V sphere equivalence for the albedo calculation is far superior to the S-equivalence and V-equivalence. These seminal papers have been very important to spread the concept of S/V sphere equivalence, and this certainly is why the snow optics community now routinely uses the SSA or d_{opt} with success to calculate albedo. SSA (or S/V) is the main geometric property driving the albedo.

However, the shape is a second driver and is the topic of this paper. In fact, the figures listed by the reviewer are in log-log scales, because, again, the goal at that time was to compare the S/V-equivalence w/r to the S-equivalence and the V-equivalence over a large range of wavelengths. Since then, Libois et al. 2013 among others showed that S/V-equivalence is not perfect, and that different ideal shapes feature slightly different co-albedo even for the same S/V (or SSA). This slight difference is directly B. Considering a spread between the extreme shapes (1.25 for spheres and 1.85 for fractals), this corresponds to a shift of 0.17 units in log10 scale used by the listed figure. This is probably less than the width of the bold and light curves in the listed figures. Nevertheless, this difference is important for applications (light penetration, albedo in climate models), now that SSA can be measured with high accuracy.

(4) In Figures 2b and 2d, the sphere is shown to have lower albedo than the other shapes. Dang et al. (2016) showed that for albedo, a model snowpack of spheres can mimic a real snowpack of non-spheres by using a smaller radius r for the sphere: the too-large g can be compensated by reducing r , which increases the SSA. Dang et al. did not examine transmittance, and the arguments in this new paper indicate that the same radius-adjustment would predict too much transmittance, as was pointed out by Libois et al. (2013). Reference 7 (Räsänen et al.) included an extensive discussion of the Dang paper.

But in our prior examination of the equivalent-sphere representation, the transmittances for clouds and snow were not systematically biased high; the errors were small and were both positive and negative of comparable magnitude, as plotted by Grenfell & Warren (1999, Figure 9) and by Neshiba et al. (2003, Figures 11, 14, 17). So I am puzzled. Any insight you can offer would be appreciated.

We agree with the reviewer that the equivalent-sphere representation works reasonably well for albedo. However the too-large asymmetry parameter (Dang et al., 2016; He and Flanner, 2020) has implications for light transmittance, as pointed out by several studies. For instance, Haussener et al. (2012) compared the transmittance of snow obtained by ray-tracing on coarse-resolution microstructure images, to the transmittance computed with DISORT (assuming equivalent-spheres). They obtained large differences (their Fig. 9b and d), the sphere yielding much more transmittance. Similar results were pointed out by Libois et al., (2013), which are

consistent with conclusions of Meirold-Mautner and Lehning (2004), who measured smaller transmittance than their equivalent-sphere model predicted.

We can not explain why Neshiba et al. (2003) obtained sometimes smaller transmittances for different shapes with respect to spheres although they systematically found a higher or equal asymmetry factor for spheres (Mie in their Figures 4-6) at wavelengths $< 1 \mu\text{m}$. This is particularly true for the equidimensional prisms (see the short wavelengths in their Fig 5 and Fig 14). Our results also agree with the exceptionally high asymmetry factor of the sphere, but we have not investigated transmittance which is of a lesser interest for snow than for clouds. Maybe this is due to the fact that the transmittance is not related in a simple way to the light penetration.

(5) Figure 2d shows that fresh snow has the lowest albedo. Commenting on this result on page 5 (six lines from the bottom), the authors say that fresh snow is a “relatively inefficient reflector”. This result is puzzling, since fresh snow probably has a larger SSA than the other snow types that were measured. Some discussion and explanation would be appreciated.

This remark has also been pointed out by reviewer #2 (specific comment #5). The reviewer is right about the fact that fresh snow is more reflective because it usually has a much higher SSA than other snow types. We agree that SSA is the main driver of the albedo, the shape comes second. In the revised manuscript, it is now clearly stated that in order to compare different snow types (and shapes) with varying sizes, the results must be interpreted at equal size/SSA (see our response to reviewer’s comment #3). In the revised manuscript we have rewritten as follows:

Interestingly, a relationship arises between the albedo and the snow type. For a given SSA, fresh snow (PP), like spheres, is a relatively inefficient reflector, [...]

(6) In Figure S4b, the median internal path length for the deformed sphere is larger than the median for the near-perfect sphere, by the factor 1.11. But for these shapes in Figure 3, B differs by a much larger factor, $1.7/1.29 = 1.32$. Why?

B is related to the mean traveled distance in the ice, and not the median. The figure was indeed misleading and has now been replaced by the new Supplementary Figure 5 (see comment #2), which better supports the new discussion about the equality $B = n^2$.

Minor comments:

Introduction, line 2. Many ice particles in clouds do not have “near-perfect geometric shapes”, as shown by the cloud-particle imager (CPI), e.g. Lawson et al. (2011).

We agree with the reviewer’s comment that ice particles in clouds can be irregular and do not have, always, a “near-perfect geometric shape”. However, hexagonally shaped ice particles can be predominant under certain conditions (mixed-phase cloud and cold temperatures), as shown by the same imager. Some halos can only be explained by the presence of such ice crystals, while other documented halos can only be explained by abnormal and more sophisticated crystal shapes (see Moilanen and Gritsevich., 2022). To be clearer, the paragraph has been amended as follows:

Ice crystals formed in the atmosphere show a large variety of sophisticated and, often, near-perfect geometric shapes (Lawson et al. 2011) (Fig. 1a). The interaction of sunlight with such crystals sometimes results into well-known optical phenomena, called halos (Greenler, 1980), whose nature is directly related to the ~~near-perfect~~ shape of the crystals (Moilanen et al., 2022).

References:

Moilanen, J. & Gritsevich, M., 2022: Light scattering by airborne ice crystals – An inventory of atmospheric halos. *Journal of Quantitative Spectroscopy and Radiative Transfer*, 290, 108313. <https://doi.org/10.1016/J.JQSRT.2022.108313>

Figure 2, 3, and 4 are hard to read because they are too small. For publication, they should be expanded, perhaps rotated so that they can run the full length of a page.

Thanks for noting this. We have slightly modified and enlarged them in the revised version, and if the manuscript is eventually accepted for publication, we will work with the editorial and production team to upgrade them to their standards.

Figure 2. Define the abbreviations PP, DF, FC, DH, RG, MF. And in line 3 of the caption, after “two-phase random medium”, insert “labelled in (a) as Malinka (2014)”.

The abbreviations correspond to the snow types referenced in the international classification of seasonal snow in the ground (Fierz et al, 2009). PP, DF, FC, DH, RG and MF stand for, respectively, **P**recipitation **P**articles, **D**ecomposing and **F**ragmented precipitation particles, **F**aceted **C**rystals, **D**epth **H**oar, **R**ounded **G**rains and **M**elt **F**orms. We have thus updated the figure caption as follows:

The dark symbols correspond to geometric shapes reported in the literature (see Supplementary Table 2) and the two-phase random medium, labeled in (a) as Malinka (2014) (see Supplementary Text 1). The colored ones correspond to the 33 natural snow samples, depending on the snow type (Fierz et al., 2009): Precipitation Particles (PP), Decomposing and Fragmented precipitation particles (DF), Faceted Crystals (FC), Depth Hoar (DH), Rounded Grains (RG) and Melt Forms (MF).

References:

Fierz, C. et al. *The International Classification for Seasonal Snow on the Ground*. (IACS Contribution N°1, UNESCO-IHP, 2009).

Page 4, last line. Cite reference 29 here.

Done.

Page 5, 5 lines from bottom. "Rounded grains nearly behave as the other idealized shapes". I think you instead mean that they behave as the other *natural* shapes, not the idealized shapes (sphere, cube, cylinder).

We actually meant *idealized* shapes, but we acknowledge that this statement was not clear enough. In the revised manuscript, we have rewritten as:

[...] while rounded grains (RG) nearly behave as ~~the~~ other idealized shapes, such as cylinders (Fig. 2b/d and Supplementary Fig. S2). [...]

Page 6 lines 3-6. This statement that spheres are useless to represent snow in radiative transfer models is too strong. Dang et al. (2016) showed how spheres can indeed represent nonspherical snow shapes for the purpose of spectral albedo, if the spherical radius is scaled.

The statement was indeed misleading. The equivalent-sphere representation is able to represent the spectral albedo of non-spherical shapes, but still fails to fully represent light penetration in snow, as already mentioned in the previous remarks. We believe that this has important consequences for the snowpack. We have rewritten as follows:

[...] Even if the spectral albedo of non-spherical shapes can be estimated using spheres by scaling their radius (Dang et al., 2016), light penetration depth in a medium with spheres is approximately twice longer than in snow with the same SSA. These results show that, in order to represent these quantities, natural snow should not be represented by the geometric shapes that have been commonly implemented in radiative transfer models, and in particular by spheres. [...]

Figure 3. It would be helpful to plot also the ratio of facet-size to wavelength. For example, for a sphere of $r=250\ \mu\text{m}$ and 10^5 facets, the radius of a facet is approximately $1.6\ \mu\text{m}$, and the ratio of facet-radius to wavelength is about 1.8.

We would like to keep the figure as simple as possible. We have added more detail in the Supplementary Fig. 3 (now Supplementary Fig. 4), where the shapes used in that part of the results are shown:

References:

Dang C., Q. Fu, and S.G. Warren, 2016: Effect of snow grain shape on snow albedo. *J. Atmos. Sci.*, 73, 3573-3583. doi: 10.1175/JAS-D-15-0276.1

Lawson, R.P., et al., 2011: Deployment of a Tethered Balloon System for Cloud Microphysics and Radiative Measurements at Ny-Ålesund and South Pole. *J. Atmos. Ocean. Technol.*, 28, 656-670.

REVIEWERS' COMMENTS

Reviewer #1 (Remarks to the Author):

The authors have properly addressed all my previous comments and questions. I really appreciate their efforts in improving the manuscript. I do not have further suggestions.

Reviewer #3 (Remarks to the Author):

The authors have adequately addressed my comments. The paper is ready for publication.

Answer to Reviewers #1 and #3

Reviewers #1 and #3 comments on **NCOMMS-23-08691A**: *Unraveling the optical shape of snow*, by Alvaro Robledano et al., Nature Communications (in review).

The reviewer's comments are reported in black, and our answers are written in blue.

Reviewer #1 (Remarks to the Author):

The authors have properly addressed all my previous comments and questions. I really appreciate their efforts in improving the manuscript. I do not have further suggestions.

We thank the reviewer for their effort in reviewing our study. We greatly appreciate their kind remarks and their help to improve the manuscript.

Reviewer #3 (Remarks to the Author):

The authors have adequately addressed my comments. The paper is ready for publication.

We thank the reviewer for their effort in reviewing our manuscript. We greatly appreciate their kind remarks and their help to improve the study.